# DIVINE: Diverse-Inconspicuous Feature Learning to Mitigate Abridge Learning

**Saheb Chhabra**                                                                       *sahebc@iiitd.ac.in*
*Department of Computer Science*
*IIIT-Delhi, New Delhi, India 110020*

**Kartik Thakral**                                                                       *thakral.1@iitj.ac.in*
*Department of Computer Science and Engineering*
*IIT Jodhpur, Rajasthan, India 342037*

**Surbhi Mittal**                                                                        *mittal.5@iitj.ac.in*
*Department of Computer Science and Engineering*
*IIT Jodhpur, Rajasthan, India 342037*

**Mayank Vatsa**                                                                         *mvatsa@iitj.ac.in*
*Department of Computer Science and Engineering*
*IIT Jodhpur, Rajasthan, India 342037*

**Richa Singh**                                                                          *richa@iitj.ac.in*
*Department of Computer Science and Engineering*
*IIT Jodhpur, Rajasthan, India 342037*

**Reviewed on OpenReview:** *https://openreview.net/forum?id=8NGKGTAD6F*

## Abstract

Deep learning algorithms aim to minimize overall error and exhibit impressive performance on test datasets across various domains. However, they often struggle with out-of-distribution (OOD) data samples. We posit that deep models primarily capture prominent features beneficial for the task while neglecting subtle yet discriminative features, a phenomenon we refer to as *Abridge Learning*. To address this issue and encourage more comprehensive feature utilization, we introduce **DIVINE** (**DIV**erse and **IN**conspicuous **FE**ature Learning), a novel approach that leverages iterative feature suppression guided by dominance maps to ensure that models engage with a diverse and complementary set of discriminative features. Through extensive experiments on multiple datasets, including MNIST, CIFAR-10, CIFAR-100, TinyImageNet, and their corrupted and perturbed variants (CIFAR-10-C/P, CIFAR-100-C/P, TinyImageNet-C/P), we demonstrate that DIVINE significantly improves model robustness and generalization. On perturbation benchmarks, DIVINE achieves mean Flip Rates (mFR) of 5.36%, 3.10%, and 21.85% on CIFAR-10-P, CIFAR-100-P, and TinyImageNet-P respectively, compared to 6.53%, 11.75%, and 31.90% for standard training methods exhibiting Abridge Learning. Moreover, DIVINE attains state-of-the-art results on CIFAR-100-P, demonstrating that addressing Abridge Learning leads to more robust models against real-world distribution variations.

## 1 Introduction

Deep learning algorithms have demonstrated remarkable success in various domains, including image classification (Taesiri et al., 2024; Thakral et al., 2024; Dehghani et al., 2023; Su et al., 2023), object detection (Wang et al., 2023b; Zong et al., 2022; Tan et al., 2020), and segmentation (Fang et al., 2023; Dosi et al., 2021; Xie et al., 2020). Despite these advancements, ensuring robustness and generalizability in real-world scenarios

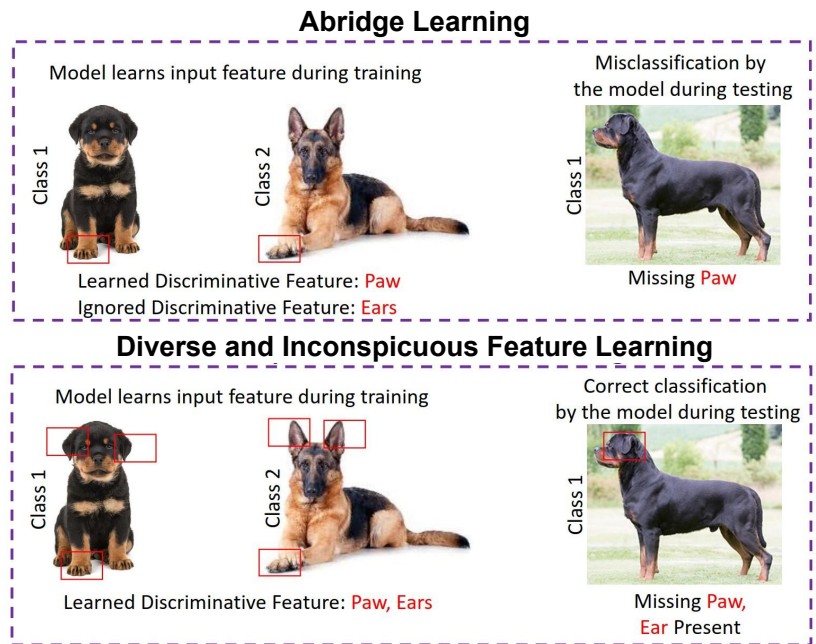

Figure 1: Illustration of *Abridge Learning* (AL) and the proposed approach. A model trained conventionally using AL learns only the 'paw' feature, neglecting other informative cues, causing failure when this feature is missing. Our proposed method, DIVINE, learns additional inconspicuous features such as the 'ear', ensuring correct classification even when dominant features are absent.

remains a significant open challenge. Current supervised deep learning paradigms primarily aim to maximize accuracy by identifying and exploiting the simplest and most prominent patterns in datasets (Geirhos et al., 2020; Keshari et al., 2019; 2018). Such an approach often results in models taking *shortcuts*, selectively focusing on dominant input features sufficient for confident classification (Dogra et al., 2024; Li et al., 2023; Ilyas et al., 2019; Pezeshki et al., 2021). Unfortunately, this leads to poor generalization performance on *out-of-distribution* data, as illustrated in Figure 1.

In practical scenarios, relying exclusively on these dominant features can lead models to become brittle under even slight variations in input data. For example, in medical image analysis, a classifier might rely excessively on the presence of medical equipment artifacts rather than subtle pathological indicators. Similarly, in autonomous driving, models may overly depend on clear road markings, neglecting less prominent yet crucial signals under adverse weather conditions. Such overreliance emphasizes the critical necessity of methods designed explicitly to encourage a comprehensive feature learning approach.

**Defining Abridge Learning:** In this research, we term this shortcut-seeking behavior as *Abridge Learning* (AL), describing a phenomenon where a model overly relies on a limited set of dominant features while ignoring other crucial yet subtle cues. Typically, these overlooked features fall into two distinct categories:

- **Inconspicuous features**: Subtle yet *causal* cues (e.g., detailed textures such as an animal's paw) that genuinely aid classification but receive small magnitude gradient signals during training.

- **Spurious features**: *Non-causal* correlations (e.g., background grass in images labeled as "cow") exploited merely due to frequent co-occurrence with class labels in training data.

Conflating these two types negatively impacts generalization: amplifying spurious correlations diminishes robustness, whereas neglecting inconspicuous features leaves crucial discriminative information underutilized. The implications of such negligence become evident in varied real-world scenarios, highlighting the necessity for explicit mechanisms to balance feature reliance effectively.

Formally, consider an observed input image $x \in \mathbb{R}^{H \times W \times C}$ generated via a Structural Causal Model (SCM):

$$x = f_X(Z), \quad y = f_Y(Z_S),$$

where $Z \in \mathbb{R}^k$ represents latent generative factors tied to semantically meaningful features, and $Z_S \subseteq Z$ indicates the subset causally influencing the label $y \in \mathcal{Y}$. Given a predictive model $f(x; \theta)$ parameterized by $\theta$, trained via empirical risk minimization using cross-entropy loss:

$$\min_{\theta} \mathbb{E}_{x,y} \left[ -y^{\top} \log f(x; \theta) \right],$$

we formally characterize Abridge Learning as the condition when the learned predictor $f(x; \theta^*)$ predominantly exploits dominant features $Z_D \subseteq Z_S$, identified by strong gradient signals and high mutual information with labels, while neglecting inconspicuous yet predictive features $Z_I = Z_S \setminus Z_D$. Mathematically, this phenomenon can be stated as:

$$I(f(x; \theta^*); Z_D) \gg I(f(x; \theta^*); Z_I), \quad \text{with} \quad I(Z_I; Y \mid Z_D) > 0.$$

In other words, the model's prediction is far more informed by dominant features ($Z_D$) than by inconspicuous ones ($Z_I$), even though the inconspicuous features still hold predictive information. Consequently, the reliance on dominant features results in significantly degraded performance under distributional shifts or feature perturbations:

$$\mathbb{E}_{x',y'}[\mathcal{L}(f(x'; \theta^*), y')] \gg \mathbb{E}_{x,y}[\mathcal{L}(f(x; \theta^*), y)],$$

where $x'$ denotes inputs with corrupted or missing dominant features. To address this challenge, we propose a novel method that explicitly suppresses overly exploited dominant signals, allowing diverse inconspicuous features to emerge and contribute meaningfully to classification. By enriching the learned representations with these subtle yet causal cues, our method significantly enhances robustness and generalization across varied data distributions.

**Research Contributions:** While existing methods aim to enhance robustness and generalizability, they typically do not explicitly focus on identifying and learning subtle yet discriminative inconspicuous features. This limitation often results in models developing shortcut-based biases, causing them to falter under real-world conditions. To bridge this gap, we introduce **DIVINE** (Diverse and Inconspicuous Feature Learning), an approach explicitly designed to mitigate shortcut reliance and facilitate the learning of a diverse set of discriminative features. Specifically, DIVINE accomplishes two primary objectives:

1. Identifying a minimal yet diverse set of inconspicuous discriminative input features (see Figure 1).

2. Training a unified model that robustly incorporates these identified inconspicuous discriminative features to enhance generalizability.

The diverse features discovered via DIVINE are explicitly disjoint, thereby maximizing representational diversity. Extensive experiments on benchmark datasets such as CIFAR10 (Krizhevsky & Hinton, 2009), MNIST (LeCun et al., 1998), CIFAR10-C (Hendrycks & Dietterich, 2019b), and CIFAR100-C (Hendrycks & Dietterich, 2019b) highlight DIVINE's effectiveness and confirm its broad applicability across multiple machine learning tasks.

## 2 Related Work

The phenomenon of *Abridge Learning* (AL), a specific form of shortcut learning where deep neural networks overly rely on dominant input features while neglecting subtle yet causal cues, poses a significant challenge

to achieving robust and generalizable models (Geirhos et al., 2020)[1]. This section reviews prior work on shortcut learning, highlighting its identification, mitigation strategies, data augmentation techniques, and causal representation learning, emphasizing their relevance to AL. We emphasize limitations in existing approaches, thereby motivating the necessity of the proposed *DIVINE* method, explicitly designed to address diverse, inconspicuous feature learning for enhanced robustness.

**Identifying Shortcut Learning:** Shortcut learning occurs when models exploit simplistic, often non-causal patterns to minimize training loss, compromising generalization on out-of-distribution data. Geirhos et al. (2020) formally conceptualized this issue, demonstrating that models trained on biased datasets tend to disproportionately focus on easily exploitable features, such as background textures, over semantically meaningful ones. For example, Carter et al. (2021) found that deep networks achieve high confidence predictions by relying only on a small subset of spurious pixels, reinforcing how standard training paradigms can inadvertently amplify non-causal correlations.

In critical domains such as medical imaging, shortcut learning has particularly severe implications. In medical contexts, Oakden-Rayner et al. (2020) showed that models trained on medical datasets often exploit irrelevant features like imaging artifacts, inflating performance metrics artificially. Such shortcuts frequently remain undetected due to evaluation metrics focused primarily on in-distribution accuracy, masking vulnerabilities in real-world deployment. Similarly, Lapuschkin et al. (2019) discussed "Clever Hans" phenomena in computer vision and gaming contexts, employing attribution heatmaps to highlight model reliance on simplistic patterns. Central to these identification issues is Gradient Starvation (Pezeshki et al., 2021), where training gradients disproportionately favor dominant features, thereby neglecting subtle yet informative cues.

**Mitigation Strategies for Shortcut Learning:** Numerous methods have addressed shortcut learning, although few explicitly target AL's unique challenges. Du et al. (2020) proposed CREX, utilizing expert-guided regularization to limit model reliance on annotated features. However, the dependency on costly annotations restricts scalability, particularly for complex datasets. DFM-X (Wang et al., 2023a) leverages prior model knowledge to guide robust feature learning, yet it requires high-quality prior models which may not always be available. Methods like SADA (Zhang et al., 2023) and DDA (Gao et al., 2023) adopt adaptive frameworks, either augmenting data strategically or using diffusion models for improved robustness, albeit indirectly addressing AL by promoting data diversity without explicitly emphasizing inconspicuous features.

Recent ensemble and gradient-based techniques offer promising directions. Carter et al. (2021) reveal that a classifier can achieve high confidence while attending to as little as 5% of spurious pixel subsets, many of which are semantically meaningless to humans. Their result substantiates our central assumption that abridge learning arises during standard training because dominant but non-causal cues monopolize the gradient signal. The Spare method (Yang et al., 2024) identifies spurious correlations via gradient analysis and importance sampling, though it may prioritize dominant signals in imbalanced scenarios. COMI (Zhao et al., 2024) integrates shortcut mitigation into empirical risk minimization, yet its efficacy can diminish with highly skewed data distributions. Addressing such imbalances, PDE (Deng et al., 2024) leverages group annotations to balance feature learning but depends critically on accurate group labels. Additionally, specialized methods targeting niche contexts, such as privacy preservation (Ling et al., 2024) and semantic segmentation (Kwon et al., 2024), often lack broader applicability.

Gradient-based regularization provides another approach to robustness enhancement. Input gradient training methods (Ross & Doshi-Velez, 2018) and Jacobian-based regularizers, including those by Varga et al. (2017) and Jacobian Adversarially Regularized Networks (JARN) (Chan et al., 2020), optimize model gradients to mitigate shortcut reliance. However, such techniques often require careful tuning and do not explicitly prioritize inconspicuous features. Moreover, existing robustness evaluations (Hoffman et al., 2019; Rusak et al., 2020; Taori et al., 2020) primarily assess performance under distributional shifts without directly addressing AL-induced feature neglect.

**Data Augmentation for Robustness** Data augmentation methods indirectly address shortcut learning by enhancing training diversity (Jin et al., 2024). CutOut (DeVries, 2017) randomly masks image regions, compelling models to utilize broader feature sets. Extensions like CutMix (Yun et al., 2019) and Mixup (Zhang

---

[1]In the literature, Shortcut Learning broadly refers to models learning unintended strategies to minimize loss. Abridge Learning is a sub-problem where models prioritize dominant cues, ignoring other relevant features.

et al., 2018) further diversify inputs through hybrid image-label constructions, albeit potentially introducing label ambiguities. AugMix (Hendrycks et al., 2019) mitigates such ambiguities by applying diverse, stochastic augmentations. Recent approaches like IPMix (Huang et al., 2023), NoisyMix (Erichson et al., 2024), DIFFUSEMIX (Islam et al., 2024a), and GenMix (Islam et al., 2024b) leverage sophisticated generative models to further enhance data diversity. While effective, these augmentations generally do not explicitly encourage learning inconspicuous features critical to addressing AL.

**Causal Representation Learning:** Causal representation learning seeks to identify and leverage semantically meaningful causal features to bolster robustness. Classical Independent Component Analysis (ICA) theory illustrates that feature identifiability typically requires stringent assumptions (Xi & Bloem-Reddy, 2023). Recent generative model studies highlight identifiability challenges in VAEs and GANs, while Lachapelle et al. (2023) proposed additive decoders to recover causal latent factors. Despite theoretical advancements, practical utility remains limited without strong priors. In contrast, our discriminative approach, DIVINE, explicitly suppresses dominant signals, facilitating the emergence and integration of previously neglected inconspicuous causal features to enhance robustness.

**Relationship to Last-Layer Ensemble (LLE):** The Last Layer Ensemble Li et al. (2023) tackles multiple known shortcuts by replacing the final classifier with a committee of heads and an auxiliary shift detector. In contrast, DIVINE pursues a fundamentally different route: it rebalances the gradient flow so that dominant and subtle features are learned simultaneously, thereby retaining accuracy even under severe corruptions and perturbations. Empirically, LLE is validated on two bespoke datasets - UrbanCars, where background and co-occurring objects act as controlled spurious cues, and ImageNet-W, which injects watermarks into images while DIVINE is stress-tested across a far broader spectrum that includes MNIST, CIFAR10/100, Tiny-ImageNet, and their corruption (-C) and perturbation (-P) suites.

**Limitations and the Need for DIVINE:** While existing methods improve robustness, they often fall short in addressing AL comprehensively. Identification techniques (e.g., Lapuschkin et al. (2019)) reveal shortcuts but do not provide actionable mitigation strategies. Mitigation methods like CREX (Du et al., 2020), LLE (Li et al., 2023), and PDE (Deng et al., 2024) rely on expert annotations, prior knowledge, or group information, which are often impractical. Data augmentation methods (e.g., AugMix, DIFFUSEMIX) enhance diversity but rarely target inconspicuous features explicitly. Causal representation learning, while theoretically promising, is constrained by identifiability challenges. In contrast, *DIVINE* explicitly suppresses dominant signals to learn diverse, inconspicuous features, offering a scalable and generalizable solution to AL without requiring prior knowledge or extensive annotations.

## 3 Diverse and Inconspicuous Feature Learning

As illustrated in Figure 2, the proposed DIVINE algorithm comprises a two-stage learning strategy designed explicitly to counteract the effects of Abridge Learning. In the first stage, DIVINE systematically identifies dominant input features using dominance feature map, which quantifies the relative importance of each input feature for classification. Once these dominant features are identified, the algorithm suppresses them, compelling the model to discover previously overlooked yet informative inconspicuous features. Subsequently, in the second stage, a unified model leverages both dominant and newly identified inconspicuous features to enhance the overall generalization performance. It is crucial to acknowledge that the exhaustive identification of all inconspicuous features is computationally intractable. Hence, DIVINE strategically identifies a diverse subset of inconspicuous features that collectively maximize model learning efficacy, thereby effectively mitigating Abridge Learning and significantly improving robustness to out-of-distribution samples.

**Illustration of Abridge Learning and Proposed Algorithm with Synthetic Data:** We conducted a synthetic experiment involving two informative features, $z_1$ and $z_2$, to explicitly demonstrate Abridge Learning. Figure 3(a) shows that under conventional training using the cross-entropy loss, the model predominantly relies on feature $z_2$, creating a horizontal decision boundary, thus clearly ignoring the predictive capability of feature $z_1$. Consequently, suppressing feature $z_2$ severely reduces model accuracy unless retrained, highlighting the pitfalls of Abridge Learning.

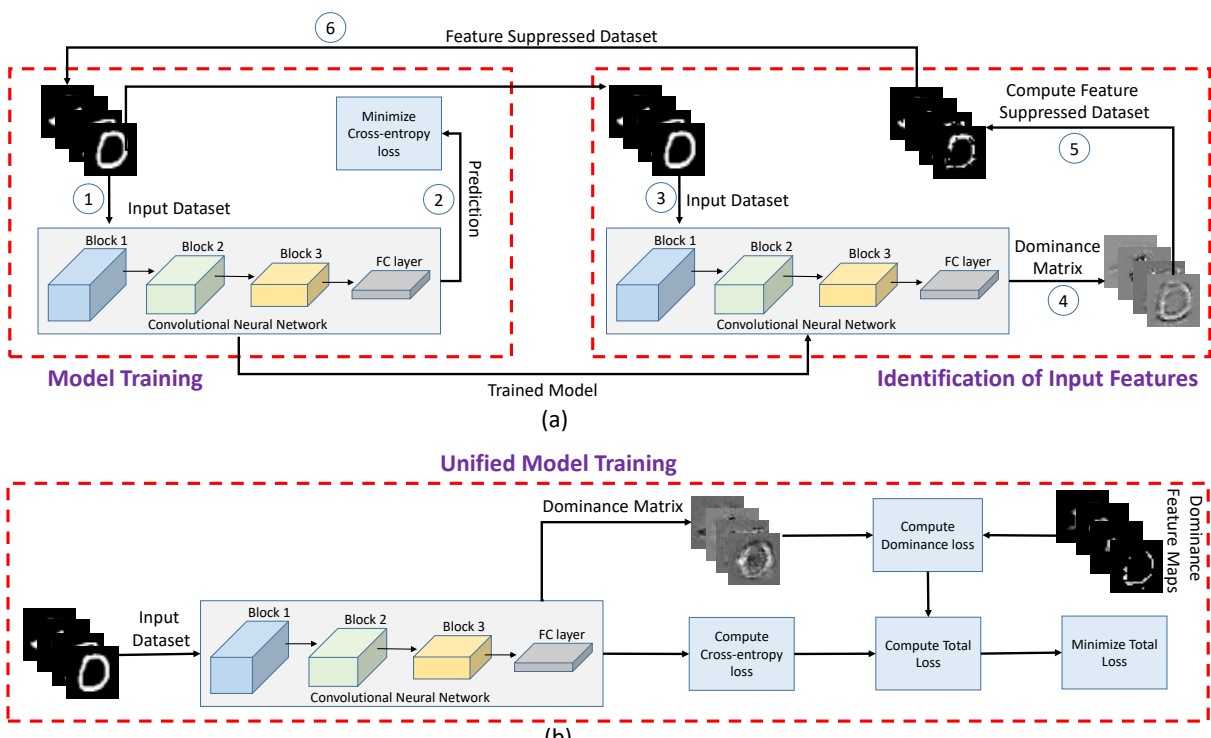

Figure 2: Pipeline of the proposed method for learning diverse and inconspicuous features. (a) Illustrates the process of identifying inconspicuous input features. Steps 1 and 2 involve training the model with original images using a cross-entropy loss function. Steps 3 and 4 depict the computation of the dominance matrix for each image. In step 5, a feature-suppressed dataset is derived from the dominance matrix and dominance feature maps. The final step involves training the model with the feature-suppressed dataset to identify inconspicuous input features. (b) Demonstrates the unified model training process with original images.

As illustrated in Figure 3(b), explicitly masking the dominant feature $z_2$ during training enables the model to learn the previously neglected feature $z_1$. This scenario confirms that $z_1$ indeed carries predictive information that was overlooked during standard training due to optimization biases favoring easily learnable dominant features. Ultimately, as depicted in Figure 3(c), the unified DIVINE model successfully integrates both $z_1$ and $z_2$ features, yielding a diagonal decision boundary reflective of balanced feature utilization. This demonstrates DIVINE's effectiveness in overcoming the gradient bias of cross-entropy training, fostering a more comprehensive and robust feature representation.

DIVINE aims to systematically identify diverse inconspicuous input features through dominance feature maps and subsequently train a unified model that robustly incorporates these features to enhance generalization performance. The remainder of this section details the feature identification method, followed by the training methodology for the unified model.

## 3.1   Identification of Dominant and Inconspicuous Input Features

As illustrated in Figure 2, the proposed DIVINE algorithm is designed explicitly to counteract Abridge Learning, a phenomenon where models overly rely on dominant input features while neglecting subtle yet predictive inconspicuous features. DIVINE systematically identifies, suppresses, and subsequently leverages these inconspicuous features through a comprehensive, two-stage learning approach, thereby significantly enhancing robustness and generalizability to out-of-distribution data. In this section, we provide the formulation of each algorithmic component.

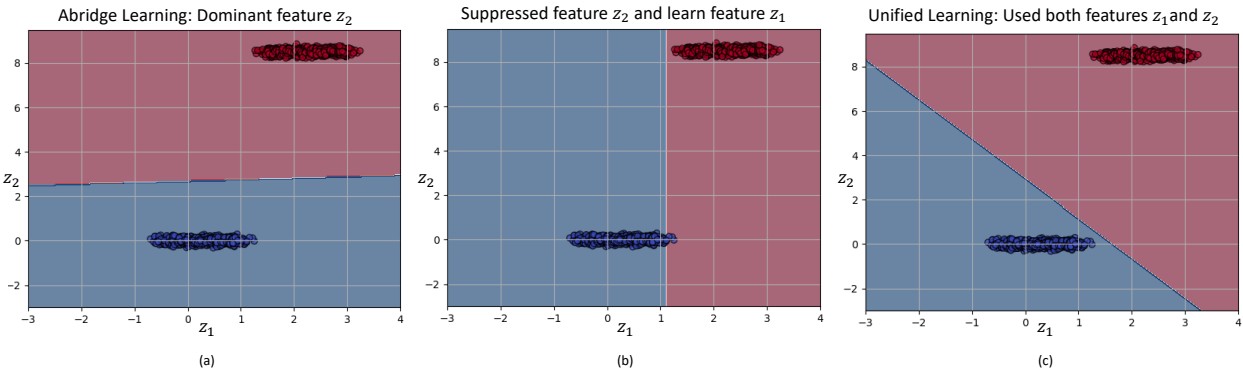

Figure 3: Illustration of Abridge Learning and the effectiveness of the proposed DIVINE on synthetic data for binary classification. (a) Under standard training, the model heavily relies on the dominant feature $z_2$, ignoring the informative yet subtle feature $z_1$. (b) Suppressing the dominant feature $z_2$ forces the model to recognize and leverage the previously ignored predictive feature $z_1$. (c) The unified learning approach of DIVINE enables the model to utilize both $z_1$ and $z_2$, resulting in improved decision-making and robustness.

We start by defining the set of input features $\mathcal{F}$, where each $F_i(x) \in \mathcal{F}$ is a masked variant of an input image $x \in \mathbb{R}^{H \times W \times C}$ with identical dimensionality $H \times W \times C$. Each $F_i(x)$ selectively highlights pixel regions representing features learned by the model. Typically, these regions correspond to subtle, inconspicuous features that standard training procedures overshadow in favor of dominant, highly salient cues. For instance, Figure 1 illustrates pixels associated with an animal's *paw* region, exemplifying an informative yet often overlooked feature.

Previous research (Geirhos et al., 2020; Pezeshki et al., 2021) indicates that traditional training methods bias models toward identifying the most dominant input features. To systematically quantify pixel dominance, DIVINE utilizes the input-output Jacobian method (Chan et al., 2020; Hoffman et al., 2019), which captures each pixel's influence on the model's predictive decisions. Formally, given an input image $x$ and a small perturbation $\epsilon$, the model output $f(x + \epsilon; \theta)$ is approximated using the Taylor series expansion around $x$:

$$f(x + \epsilon; \theta) = f(x; \theta) + \epsilon \frac{df(x; \theta)}{dx} + \mathcal{O}(\epsilon^2). \tag{1}$$

By neglecting higher-order terms for small perturbations, we simplify the expression to:

$$f(x + \epsilon; \theta) \approx f(x; \theta) + \epsilon \frac{df(x; \theta)}{dx}, \tag{2}$$

where the input-output Jacobian matrix is represented by $\frac{df(x;\theta)}{dx}$. Since we specifically compute Jacobians concerning the true class prediction, we denote this as the Dominance matrix $D_1(x) \in \mathbb{R}^{H \times W \times C}$:

$$D_1(x) = \frac{df(x; \theta)}{dx}. \tag{3}$$

In the dominance matrix $D_1(x)$, larger magnitude values (positive or negative) indicate pixels with substantial influence over the model's prediction, thus marking them as dominant pixels.

To identify the first dominant input feature, we select the top $p\%$ of pixels with the highest absolute dominance values. A binary mask $M_1(x) \in \mathbb{R}^{H \times W \times C}$ is defined based on a threshold $t$, computed by sorting dominance values:

$$M_1(x) = \begin{cases} 1, & \text{if } |D_1(x)| \geq t \\ 0, & \text{otherwise} \end{cases}. \tag{4}$$

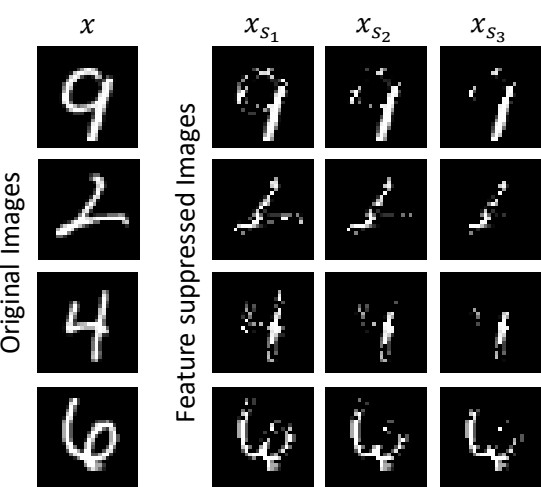

Figure 4: Sample of original images and the corresponding intermediate feature-suppressed images from the MNIST dataset obtained using the proposed method.

The first identified feature $F_1(x)$ is then obtained by element-wise multiplication of the mask with the original image:

$$F_1(x) = M_1(x) \odot x. \tag{5}$$

Subsequently, a corresponding dominance feature map $D_{m_1}(x)$ explicitly capturing the dominance values of $F_1(x)$ is computed as:

$$D_{m_1}(x) = M_1(x) \odot D_1(x). \tag{6}$$

### 3.2 Iterative Suppression and Discovery of Diverse Inconspicuous Features

To systematically uncover additional inconspicuous features and ensure feature diversity, DIVINE explicitly enforces that identified features are mutually disjoint, formally represented as:

$$F_i(x) \cap F_j(x) = \phi, \quad \forall i \neq j. \tag{7}$$

Achieving disjointness requires suppressing the previously identified dominant features. We thus suppress the initially identified dominant feature $F_1(x)$ by setting the corresponding pixels to zero. This suppression is applied pixel-wise as follows:

$$x_{s_1} = \begin{cases} x, & \text{if } |D_{m_1}(x)| = 0 \\ 0, & \text{if } |D_{m_1}(x)| \neq 0 \end{cases}, \tag{8}$$

resulting in a suppressed dataset $\mathbf{X}_{s_1}$ created by applying this suppression step to all images in the original dataset $\mathbf{X}$. By training a new model on $\mathbf{X}_{s_1}$, recalculating dominance values, and repeating suppression iteratively, DIVINE progressively identifies further inconspicuous features. Figure 4 visually illustrates original and suppressed MNIST images, clearly demonstrating how suppressing dominant features reveals inconspicuous yet informative regions. This iterative process continues until a predefined stopping criterion is met (see Subsection 4.3), usually involving diminishing returns in feature novelty or computational limits.

### 3.3 Unified Model Training Incorporating Identified Features

Upon identifying a comprehensive set of diverse inconspicuous features, DIVINE advances to training a unified model explicitly designed to leverage all discovered features simultaneously. Suppose $k$ inconspicuous input features are identified for an input image $x$, each associated with its dominance feature map

$D_{m_1}(x), D_{m_2}(x), \ldots, D_{m_k}(x)$. These individual feature maps are combined into a unified dominance map $D_m(x)$ through summation:

$$D_m(x) = \sum_{i=1}^{k} D_{m_i}(x). \tag{9}$$

The unified model $f_u(x; \theta)$, parameterized by $\theta$, is then trained using the original dataset $\mathbf{X}$ to comprehensively leverage all the identified features. Training employs a composite loss function composed of standard cross-entropy and a dominance loss to promote balanced feature utilization:

$$\min_{\theta} \; \mathbb{E}_{(x,y)} \left[ -y^T \log f_u(x; \theta) + ||D_m(x) - D_u(x)||_2^2 \right], \tag{10}$$

where $D_u(x)$ denotes the dominance matrix computed by the unified model $f_u(x; \theta)$. The dominance loss specifically aligns the unified dominance feature map $D_m(x)$ with the learned dominance values ($D_u(x)$) of the model:

$$L_D(x) = ||D_m(x) - D_u(x)||_2^2. \tag{11}$$

Optimizing this composite loss explicitly counteracts the bias introduced by traditional cross-entropy training, which predominantly favors dominant features. Consequently, DIVINE's unified model robustly integrates both previously dominant and newly identified inconspicuous features, effectively overcoming the limitations posed by Abridge Learning. As a result, DIVINE significantly enhances model robustness and generalization, especially under challenging out-of-distribution conditions.

## 4 Experimental Setup

The primary objective of this paper is to remove the *shortcuts* learned by the model via learning inconspicuous and diverse input features. Our hypothesis is based on the observation that existing algorithms learn dominant features while ignoring other relevant features from the dataset within a distribution. The performance of the models suffers when dominant features are distorted/suppressed. This is because the inductive bias of the model is based on the dominant feature only. To address this, the proposed DIVINE algorithm is designed to learn the dominant features along with inconspicuous features reducing the dependence on the dominant feature only. Since the proposed learning process introduces the model to suppressed features as well, making the training diverse in nature. This ensures the generalized inductive bias of the final trained model. In order to validate this hypothesis, the experiments are performed for Abridged Learning on the feature-suppressed datasets corresponding to MNIST (LeCun et al., 1998), CIFAR10, CIFAR100 (Krizhevsky & Hinton, 2009), and TinyImageNet (Le & Yang, 2015) datasets. These feature-suppressed datasets are obtained by suppressing the identified input features (described in Section 3.1).

To further analyze the applicability of the proposed algorithm in real-world scenarios, we perform the experiment by evaluating the unified model on out-of-distribution samples. These samples are taken from the corrupted datasets CIFAR10-C, CIFAR100-C, TinyImageNet-C and perturbed datasets CIFAR10-P, CIFAR100-P, and TinyImageNet-P. The corrupted datasets contain 15 different corruptions corresponding to the CIFAR10, CIFAR100, and TinyImageNet datasets, respectively. We discuss the details of the corruption and perturbed datasets employed for evaluation below:

- **MNIST (LeCun et al., 1998)** consists of 60,000 training images and 10,000 testing images, of handwritten digits from 10 different classes (0-9). Each image is a grayscale image of $28 \times 28$ resolution. The standard pre-defined protocol is used for evaluation.

- **CIFAR10 (Krizhevsky & Hinton, 2009)** contains 60,000 $32 \times 32$ color images of 10 different classes with 50,000 images in training set and 10,000 images in testing set. The standard pre-defined protocol is used for evaluation.

- **CIFAR100 (Krizhevsky & Hinton, 2009)** contains 60,000 $32 \times 32$ color images across 100 fine-grained classes grouped into 20 superclasses with 50,000 images in training set and 10,000 images in testing set. The standard pre-defined protocol is used for evaluation.

- **TinyImageNet (Le & Yang, 2015)** is a large-scale dataset consisting of 100,000 images of 200 classes (500 for each class). The dataset consists of color images of $64 \times 64$ resolution. The standard pre-defined protocol is used for evaluation.

- **CIFAR10-C (Hendrycks & Dietterich, 2019b)** contains 15 corruptions on the CIFAR10 dataset - defocus blur, contrast, pixelate, snow, fog, glass blur, brightness, elastic transform, frost, jpeg compression, shot noise, impulse noise, zoom blur, Gaussian noise, and motion blur. The images are corrupted at five different levels with increasing severity.

- **CIFAR100-C (Hendrycks & Dietterich, 2019b)** contains 15 corruptions on the CIFAR100 dataset. The corruptions used on the CIFAR10-C dataset are also used on this dataset. The images are corrupted at five different levels with increasing severity.

- **TinyImageNet-C (Hendrycks & Dietterich, 2019a)** contains 15 corruptions on the TinyImageNet dataset. The corruptions used in CIFAR10 and CIFAR100 datasets and also used in this dataset. The images are corrupted at five different levels with increasing severity. This results in a total of 75 distinct corruptions.

- **CIFAR10-P, CIFAR100-P, and TinyImageNet-P (Hendrycks et al., 2019):** The perturbed datasets CIFAR10-P, CIFAR100-P, and TinyImageNet-P modify the original CIFAR and ImageNet datasets. These datasets have smaller perturbations compared to corruption datasets and are used to measure the model's prediction stability. Each example in these datasets is a video and we measure the model's prediction consistency. Ideally, the model should not change the prediction with the increase in perturbation intensity.

### 4.1 Comparison with Existing Approaches

To thoroughly evaluate the effectiveness of DIVINE, we provide comparisons against relevant alternative approaches that involve input pixel manipulation or gradient-based regularization. Since the DIVINE algorithm systematically suppresses dominant input pixels based on dominance maps, we include a *Random Suppression (RS)* baseline. In RS, a random subset comprising $p\%$ of input pixels is suppressed (set to zero) during training, allowing us to determine whether the systematic selection via dominance maps provides a significant advantage over random pixel suppression.

Additionally, we compare DIVINE with the established Jacobian Regularization method from prior literature (Chan et al., 2020). Jacobian Regularization is a widely utilized approach for enhancing model robustness by reducing sensitivity to small input perturbations. Specifically, this method applies regularization to the input-output Jacobian, encouraging smoother decision boundaries and mitigating model sensitivity to specific input pixels. We implement Jacobian Regularization combined with the standard cross-entropy loss to provide a meaningful baseline for assessing the benefits of DIVINE's structured approach to dominance-based feature suppression.

**Rationale for Not Comparing DIVINE with Data Augmentation Methods on Corrupted Datasets:** In existing literature, various data augmentation techniques such as AugMix (Hendrycks et al., 2019), CutMix (Yun et al., 2019), and Mixup (Zhang et al., 2018) have been proposed to enhance model robustness against distribution shifts, particularly when evaluated on corrupted datasets such as CIFAR10-C, CIFAR100-C, and TinyImageNet-C. Although DIVINE is evaluated using these corrupted datasets, direct comparison to these data augmentation methods is not appropriate for several fundamental reasons:

1. **Different Underlying Objectives**: DIVINE explicitly addresses the model's inherent bias toward dominant input features by iteratively identifying, suppressing, and promoting the learning of inconspicuous features. It modifies the model's intrinsic learning dynamics based on structured supervision via dominance matrices. Conversely, data augmentation methods primarily modify input data to increase diversity without explicitly correcting model biases or systematically influencing the internal feature-learning processes.

Table 1: Comparison of classification accuracy (%) of existing algorithms and the DIVINE on the original and feature-suppressed datasets.

| | Abridge Learning | Jacobian Regularization | Random Suppression | DIVINE |
|---|---|---|---|---|
| **MNIST** | | | | |
| **Original** | 99.21 | 85.80 | 98.67 | 97.31 |
| $\mathbf{X}_{s_1}$ | 66.53 | 76.52 | 85.56 | 91.41 |
| $\mathbf{X}_{s_2}$ | 50.98 | 70.18 | 68.10 | 84.98 |
| $\mathbf{X}_{s_3}$ | 40.87 | 62.97 | 54.24 | 75.20 |
| **Average** | 64.40 | 73.87 | 71.64 | 87.22 |
| **CIFAR10** | | | | |
| **Original** | 82.38 | 81.63 | 81.73 | 80.34 |
| $\mathbf{X}_{s_1}$ | 64.16 | 65.90 | 63.97 | 67.41 |
| $\mathbf{X}_{s_2}$ | 52.63 | 54.88 | 52.01 | 58.18 |
| $\mathbf{X}_{s_3}$ | 44.45 | 46.87 | 45.19 | 51.83 |
| **Average** | 60.98 | 62.32 | 60.72 | 64.44 |
| **CIFAR100** | | | | |
| **Original** | 74.63 | 75.95 | 75.46 | 74.57 |
| $\mathbf{X}_{s_1}$ | 37.44 | 48.31 | 45.21 | 56.31 |
| $\mathbf{X}_{s_2}$ | 27.84 | 39.61 | 36.86 | 49.79 |
| $\mathbf{X}_{s_3}$ | 15.22 | 26.28 | 14.71 | 33.49 |
| **Average** | 39.23 | 44.64 | 43.06 | 53.54 |
| **Tiny-ImageNet** | | | | |
| **Original** | 60.59 | 49.06 | 51.66 | 54.59 |
| $\mathbf{X}_{s_1}$ | 40.46 | 39.29 | 43.71 | 43.42 |
| $\mathbf{X}_{s_2}$ | 27.35 | 31.91 | 33.92 | 32.67 |
| $\mathbf{X}_{s_3}$ | 21.67 | 27.28 | 27.94 | 26.66 |
| **Average** | 37.51 | 36.88 | 39.30 | 39.33 |

2. **Distinct Methodological Paradigms**: DIVINE directly intervenes in the feature learning process to mitigate biases and enhance robustness. Data augmentation approaches, however, assume that generalization improvements arise indirectly through increased data variability and do not directly address model biases related to feature dominance or neglect.

3. **Complementarity Rather than Competition**: Positioning DIVINE directly against data augmentation methods could lead to misinterpretation, suggesting competition rather than complementarity. DIVINE and data augmentation serve fundamentally distinct but complementary purposes. DIVINE focuses on explicitly correcting feature-learning biases, while data augmentation aims to enrich dataset diversity. Notably, integrating data augmentation with DIVINE may further amplify robustness, underscoring the potential synergies rather than competition between these approaches.

Hence, the scope of our evaluations and comparisons intentionally excludes direct benchmarking against data augmentation techniques, focusing instead on methods that directly influence the feature-learning process or pixel-level manipulation through explicit mechanisms.

## 4.2 Evaluation Metrics

We compute the classification accuracy to evaluate model performance on the MNIST and CIFAR10 datasets. To evaluate the performance on the corrupted images, we report the Relative Corruption Error (Relative CE) and Relative Mean Corruption Error (Relative mCE) (Hendrycks & Dietterich, 2018).

$$\text{Relative } CE_c^f = \frac{\sum_{s=1}^{5} E_{s,c}^f - E_{original}^f}{\sum_{s=1}^{5} E_{s,c}^b - E_{original}^b} \tag{12}$$

where, $f$ denotes the model to be evaluated, $b$ denotes the baseline model obtained using Abridge Learning (AL), $E_{clean}^f$ and $E_{clean}^b$ denote the error obtained corresponding to the model to be evaluated and the AL model on original images. $E_{s,c}^f$ and $E_{s,c}^b$ denote error rates on corruption $c$ at severity level $s$ corresponding

to the model to be evaluated and the AL model, respectively. A lower Relative CE indicates a higher performance over the baseline.

To evaluate the performance on the perturbed datasets, we measure the probability that two consecutive frames with different intensity of perturbations, have "flipped" or mismatched predictions. This is termed as mean Flip Rate (mFR) (Hendrycks et al., 2019).

### 4.3 Selection Criteria for Number of Feature Maps

In order to decide the number of features, we have computed the running average of the classification accuracy obtained on the original and feature suppressed dataset using AL method. We decided to iterate computing feature maps at most 3 times given the average running classification should not be below 50% of the classification accuracy obtained on the original dataset.

### 4.4 Implementation Details

For the MNIST dataset, a Convolutional Neural Network (CNN) architecture is used. The network consists of 5 convolutional layers, each followed by ReLU activation and maxpool. Two fully connected layers of dimensions 512 and 64 are added after the convolutional block followed by softmax. ResNet50, XceptionNet, and DenseNet121 architectures are used for the CIFAR10, CIFAR100, and TinyImageNet datasets. Models are trained on the original and feature-suppressed datasets with a learning rate of 0.0001. For the CIFAR10 and CIFAR100 datasets, models are trained using Adam optimizer with a batch size of 32 whereas, models are trained using RMSProp optimizer with a batch size of 64 for the TinyImageNet dataset. For the MNIST dataset, the models are trained for 10 epochs and the features are suppressed with $p = 3\%$. The models on the CIFAR10 and CIFAR100 datasets are trained for 20 epochs and $p = 3\%$ is used.

For both datasets, the architectures used to train the AL models are used for training the unified model. The unified models are trained using Adam optimizer with a learning rate of 0.0001 for 20 epochs and batch size of 32. Code is implemented in Tensorflow 2.3.1. The model trainings are performed on a DGX station with Intel Xeon CPU, 256 GB RAM, and four 32 GB Nvidia V100 GPU cards. The source code is available at `https://github.com/Saheb-Chhabra/ID-Learning/`.

## 5 Results and Analysis

The proposed DIVINE algorithm is evaluated on two types of datasets: (i) feature-suppressed datasets and (ii) corruptions. In the first set of evaluations, we validate our assertion of "Abridge Learning" using the MNIST, CIFAR10, CIFAR100, and TinyImageNet datasets. For corruptions, we showcase results on CIFAR10-C, CIFAR100-C, and TinyImageNet-C datasets. We further compare the DIVINE algorithm with an algorithm proposed by Carter et al. (2021).

### 5.1 Evaluation on Feature-suppressed Datasets

This experiment validates that conventionally trained models rely on dominant features, causing performance to degrade when those features are absent. We evaluate the performance of the AL model trained on original images using both original and feature-suppressed test sets, i.e., $\mathbf{X}$, $\mathbf{X}_{s_1}$, $\mathbf{X}_{s_2}$ and $\mathbf{X}_{s_3}$. Dataset $\mathbf{X}_{s_1}$ has

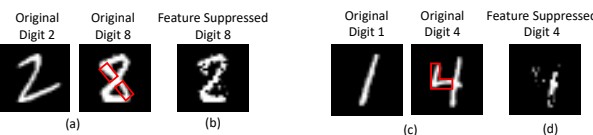

Figure 5: Visualizations of semantically relevant features learned by the model. (a) shows the strokes learned by the model, which distinguishes digit 8 from 2, and (b) shows the feature-suppressed image. Similarly, (c) and (d) show the distinguishing strokes learned by the model and the feature-suppressed image.

Table 2: Comparison of relative corruption error obtained using the existing and proposed methods for each corruption type on the CIFAR10-C and CIFAR100-C datasets.

| Corruptions | CIFAR10-C | | | CIFAR100-C | | |
|---|---|---|---|---|---|---|
| | Jacobian Regularization | Random Suppression | DIVINE | Jacobian Regularization | Random Suppression | DIVINE |
| Defocus Blur | 102.35 | 108.77 | 92.90 | 105.88 | 108.38 | 102.08 |
| Contrast | 100.19 | 103.55 | 99.64 | 103.86 | 114.32 | 122.66 |
| Pixelate | 95.09 | 113.08 | 75.58 | 108.59 | 106.95 | 74.61 |
| Snow | 91.57 | 81.15 | 81.38 | 97.17 | 99.32 | 78.91 |
| Fog | 105.04 | 111.62 | 99.17 | 111.01 | 110.61 | 110.52 |
| Glass Blur | 96.11 | 87.62 | 73.69 | 98.51 | 98.49 | 82.99 |
| Brightness | 106.40 | 133.70 | 101.00 | 129.75 | 136.48 | 125.60 |
| Elastic | 103.84 | 109.48 | 97.95 | 107.35 | 106.55 | 96.89 |
| Frost | 100.47 | 113.05 | 88.12 | 104.53 | 111.80 | 83.74 |
| JPEG | 86.40 | 93.95 | 70.86 | 93.23 | 91.82 | 73.24 |
| Shot Noise | 95.65 | 111.77 | 72.62 | 101.47 | 107.27 | 76.46 |
| Impulse Noise | 91.70 | 92.17 | 73.02 | 105.16 | 96.99 | 78.15 |
| Zoom Blur | 99.16 | 111.06 | 95.85 | 97.47 | 98.42 | 89.50 |
| Gaussian Noise | 95.49 | 110.70 | 73.32 | 101.74 | 107.97 | 79.18 |
| Motion Blur | 105.37 | 116.31 | 99.40 | 107.86 | 113.26 | 98.38 |
| **Relative mCE** | 98.32 | 106.29 | **86.30** | 104.90 | 107.24 | **91.53** |

Table 3: Classification accuracy (%) obtained using Jacobian Regularization (JR), Random Suppression (RS), and DIVINE algorithm on the TinyImageNet-C dataset for different corruptions.

| Corruptions | Jacobian Regularization | Random Supression | DIVINE |
|---|---|---|---|
| Defocus Blur | 21.07 | 19.97 | 29.43 |
| Contrast | 12.37 | 12.51 | 18.70 |
| Pixelate | 27.98 | 31.79 | 39.53 |
| Snow | 25.16 | 26.83 | 32.80 |
| Fog | 21.46 | 23.37 | 33.09 |
| Glass Blur | 28.33 | 30.36 | 32.14 |
| Brightness | 27.02 | 28.33 | 36.14 |
| Elastic | 21.23 | 21.47 | 30.89 |
| JPEG | 26.30 | 29.07 | 37.25 |
| Shot Noise | 35.77 | 37.68 | 43.09 |
| Impluse Noise | 29.71 | 34.09 | 39.14 |
| Zoom Blur | 20.07 | 18.83 | 28.20 |
| Gaussian Noise | 34.75 | 36.99 | 41.84 |
| Motion Blur | 21.47 | 22.11 | 30.87 |
| **Mean** | 25.19 | 26.67 | **33.79** |

images with one suppressed dominant input feature in each image. Similarly, datasets $\mathbf{X}_{s_2}$ and $\mathbf{X}_{s_3}$ have images with two and three suppressed dominant input features in each image, respectively. Table 1 shows the performance of the AL models corresponding to the MNIST, CIFAR10, CIFAR100, and TinyImageNet datasets. It is observed that the performance of the AL models degrades significantly on the feature-suppressed datasets. For instance, the performance of the AL model trained on the MNIST dataset drops from 99.21% to 66.53% on feature-suppressed dataset $\mathbf{X}_{s_1}$ (32.68% drop), which further degrades to 50.98% on feature-suppressed dataset $\mathbf{X}_{s_2}$. This shows that the performance of the models trained using conventional methods is heavily dependent on the dominant input features. On the other hand, the performance of the unified model drops by only 5.90% when evaluated on $\mathbf{X}_{s_1}$. As seen in Table 1, the unified model trained using the proposed DIVINE algorithm performs well on the feature-suppressed datasets.

Results of the unified model in Table 1 are compared with random suppression, and the Jacobian regularization method (Chan et al., 2020). Both approaches are used to enhance the robustness of the models. It is observed that existing approaches perform better than the AL model, especially on the MNIST dataset. However, the performance is not as good as that obtained using the unified model. In the random suppression method, there is no supervision to the model for learning a diverse set of features. While in the Jacobian

regularization method, the model reduces its dependency on the dominant features during training, it is not able to learn the inconspicuous features.

Figure 5 (a) & (b), show that the discriminative stroke of digit '8' (highlighted in red) is the most dominant feature differentiating it from digit '2', and is therefore suppressed. Similar observations can be made for digits '4' and '1' in Figure 5 ((c) & (d)). Figure 6 effectively illustrates the core principle of the DIVINE algorithm: iterative feature suppression forces the model to learn inconspicuous, diverse features that enhance generalization and robustness. The key insights from the Figure 6 are as follows:

1. The original images contain all features, including dominant and inconspicuous ones, which the model initially uses for classification.

2. Feature-suppressed images represent the results of successive iterations of feature suppression. Considering the first row representing the sample of a class horse:

    (a) The image $x_{s_1}$ represents the suppression of the dominant feature identified by the model specifically the horse's body.

    (b) After suppressing the first dominant feature, the model identifies and suppresses the second dominant feature i.e., parts of the horse's legs, as shown in $x_{s_2}$.

    (c) In the third iteration, the next set of dominant features is suppressed, which are other structural elements like part of the tail or mane, as shown in $x_{s_3}$.

3. The features learned by the model during suppression are semantically meaningful. For example: The model's attention transitions from the horse's body and legs (dominant features) to the tail, mane, and even background details (inconspicuous features). This process ensures that the model does not overly depend on any single set of dominant features for classification.

The visualization of the dominance matrix computed corresponding to both the AL and unified models is shown in Figure 7. We have made the following observations from Figure 7:

1. The dominance matrices $D_1(x)$ corresponding to AL models are highly concentrated on specific dominant input features. This confirms that AL models primarily rely on dominant features for making predictions, neglecting other potentially useful inconspicuous features.

2. The matrices $D_2(x)$ and $D_3(x)$, which correspond to models trained on feature-suppressed datasets, show a shift in focus from dominant features to inconspicuous features.

3. The unified dominance matrix $D_u(x)$, representing the fully trained DIVINE model, shows a balanced focus across all identified input features. This includes both dominant and inconspicuous features, ensuring robustness and generalization.

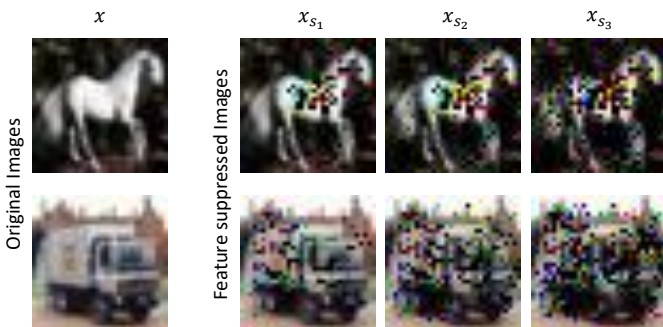

Figure 6: A few sample original images and the corresponding intermediate feature-suppressed images on the CIFAR10 dataset obtained using the proposed method.

Table 4: mean Flip Rate (mFR) obtained using Abridged Learning and DIVINE algorithm on the CIFAR10-P and CIFAR100-P datasets for different perturbations. A lower mFR indicates better performance.

| mFR % | CIFAR10 | | CIFAR100 | |
|---|---|---|---|---|
| | Abridged Learning | DIVINE | Abridged Learning | DIVINE |
| Brightness | 1.33 | 1.22 | 2.98 | 1.03 |
| Gaussian Noise | 5.21 | 3.86 | 15.30 | 2.21 |
| Motion Blur | 11.26 | 9.90 | 14.74 | 2.10 |
| Rotate | 8.29 | 6.24 | 11.12 | 3.09 |
| Scale | 9.55 | 7.99 | 13.20 | 4.87 |
| Shot Noise | 6.40 | 4.82 | 17.96 | 2.90 |
| Snow | 3.75 | 2.96 | 6.59 | 1.15 |
| Tilt | 3.13 | 2.49 | 5.52 | 1.55 |
| Translate | 15.63 | 13.53 | 28.34 | 11.79 |
| Zoom Blur | 0.79 | 0.67 | 1.78 | 0.32 |
| Overall mFR | 6.53 | **5.36** | 11.75 | **3.10** |

Figure 7: Illustration of the dominant and inconspicuous features obtained in the MNIST dataset. Sample original images $x$ and their corresponding dominance matrices are shown. $D_1(x)$ and $D_u(x)$ are obtained from the Abridge Learning (AL) and unified DIVINE models, respectively, trained on the original dataset. In contrast, $D_2(x)$ and $D_3(x)$ are obtained from models trained on feature-suppressed datasets. Note the progression from the highly concentrated features in $D_1(x)$ to the more distributed and balanced feature focus in $D_u(x)$, which visually confirms that the DIVINE model learns a comprehensive set of features.

4. The broader focus in $D_u(x)$ aligns with the improved performance of DIVINE on feature-suppressed datasets (as highlighted in Table 1) and on out-of-distribution datasets.

Overall, Figure 7 provides a clear visualization of the progression from narrow feature focus (AL) to comprehensive feature learning (DIVINE). The shift in dominance matrices underscores the success of DIVINE in mitigating Abridge Learning by promoting the learning of diverse and inconspicuous features. This balanced learning enhances robustness and generalization, as demonstrated in the experimental results.

We have also validated the above learning dynamics on the CIFAR10 dataset using Grad-CAM visualizations in Figure 8. The visualizations in Figure 8 qualitatively demonstrate that the proposed DIVINE model attends to more semantically meaningful and diverse regions of the input compared to the Abridge Learning (AL) model. While AL often fixates on narrow or spurious cues such as textures or localized patches, DIVINE distributes attention over broader, object-centric regions. This indicates that AL models tend to over-rely on dominant features and overlook other predictive cues, leading to limited generalization. In contrast, DIVINE effectively suppresses such dominance and recovers inconspicuous yet informative features, resulting in improved interpretability and robustness across examples.

**Ablation Study for parameter $p$:** On updating the values of $p$ from 3% to 10%, the performance of the unified model degrades on feature-suppressed datasets. Since, the model prediction is dependent only on a few input pixels, setting a higher value of $p$ results in suppressing of dominant as well as other input features, which in turn decreases the performance of the unified model on the feature-suppressed datasets.

## 5.2 Evaluation on Corruptions and Perturbations

This experiment is performed to evaluate the generalizability and robustness of the proposed unified model on out-of-distribution images. The performance on corruptions and perturbations are discussed below:

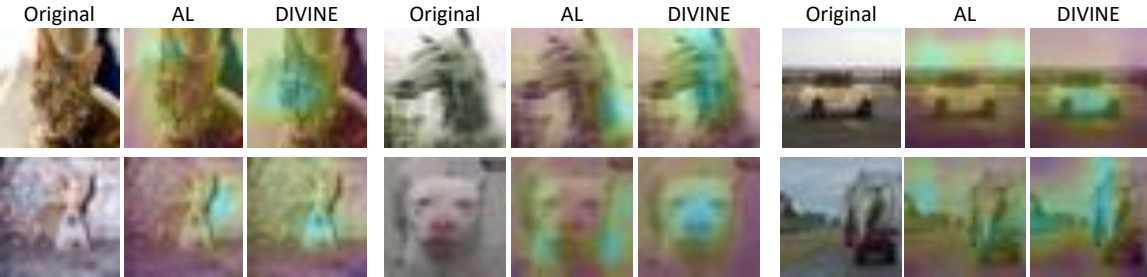

Figure 8: Grad-CAM visualizations comparing attention regions of models trained using standard Abridge Learning (AL) and proposed DIVINE approach on the CIFAR10 dataset. The leftmost column in each triplet shows the original input image, the middle column shows Grad-CAM maps for the AL model, and the rightmost column shows Grad-CAM maps for the DIVINE model. DIVINE attends to more semantically diverse, object-centric regions, while AL focuses narrowly on dominant or spurious cues.

**Performance on Corruption** We have reported the relative corruption error (Relative CE) and the relative mean corruption error (Relative mCE) on the CIFAR10-C and CIFAR100-C datasets. The results are shown in Table 2. The proposed unified model outperforms random suppression and Jacobian Regularization methods on the corruption datasets. The proposed unified model gives a Relative mCE of **86.30** and **91.53** corresponding to the CIFAR10-C and CIFAR100-C datasets, respectively. On comparing the performance of CIFAR10-C and CIFAR100-C datasets for individual corruptions, the proposed unified model trained with the DIVINE method outperforms other methods on 14 corruptions (excluding snow corruption on the CIFAR10-C and contrast corruption on the CIFAR100-C) and gives a comparable performance on the snow and contrast corruption corresponding to CIFAR10-C dataset, respectively.

Table 1 shows that the performance of existing methods is comparable to the original images. However, random suppression and Jacobian regularization methods fail to generalize well on out-of-distribution images as shown in Table 2 due to the following reasons:

1. In case of random suppression, the model lacks supervision to focus on meaningful inconspicuous features. This approach does not ensure that suppressed features are either dominant or relevant, potentially leading to suboptimal learning.

2. In case of Jacobian Regularization (JR), it effectively reduces dependency on dominant features, however, it does not actively guide the model to learn inconspicuous features, leaving a gap in robust feature learning.

To test our method on a large-scale dataset, we computed the performance on the TinyImageNet-C dataset and report the results obtained in Table 3. On the TinyImageNet-C dataset, DIVINE yields an absolute improvement of 25.54% and 19.02% over the random suppression and Jacobian regularization methods, respectively as shown in Table 3. We observe the robustness of the proposed DIVINE algorithm against a variety of corruptions. These results are illustrated for three different learning methods namely- Jacobian Regularization, Random Suppression, and the proposed DIVINE algorithm. From the table, it is clearly visible that the proposed algorithm outperforms other algorithms on all corruptions. We also achieve significantly better mean classification accuracy using the proposed DIVINE algorithm. This clearly describes the applicability of DIVINE algorithm on large-scale datasets as well.

**Performance on Perturbations** We have reported the Flip Rate (FR) and the Overall mean flip rate (Overall mFR) on the CIFAR10-P, CIFAR100-P, and TineImageNet-P datasets. The results are shown in Tables 4 and 5. The proposed unified model outperforms abridge learning on the perturbed datasets. The proposed unified model gives an overall mFR of **5.36%**, **3.10%**, and **21.85%** corresponding to the CIFAR10-P, CIFAR100-P, and TinyImageNet-P datasets, respectively. On comparing the performance of individual perturbations on all datasets, the proposed unified model trained with the DIVINE method outperforms

Table 5: mean Flip Rate (mFR) obtained using Abridged Learning and DIVINE algorithm on the TinyImageNet-P dataset for different perturbations. A lower mFR indicates better performance.

| mFR % | TinyImageNet | |
|---|---|---|
| | Abridged Learning | DIVINE |
| Brightness | 12.09 | 9.53 |
| Gaussian Noise | 28.56 | 18.58 |
| Gaussian Noise V3 | 56.82 | 42.36 |
| Rotate | 45.49 | 30.19 |
| Shear | 37.83 | 24.19 |
| Shot Noise V2 | 41.11 | 26.99 |
| Snow | 15.66 | 11.26 |
| Speckle Noise | 26.40 | 15.26 |
| Speckle Noise V3 | 53.97 | 38.08 |
| Translate | 41.92 | 31.51 |
| Gaussian Blur | 5.17 | 3.91 |
| Gaussian Noise V2 | 47.06 | 32.98 |
| Motion blur | 7.66 | 5.36 |
| Scale | 40.3 | 29.87 |
| Shot Noise | 32.09 | 20.14 |
| Shot Noise V3 | 50.29 | 34.85 |
| Spatter | 17.50 | 12.22 |
| Speckle Noise V2 | 44.10 | 28.65 |
| Tilt | 25.26 | 15.45 |
| Zoom Blur | 8.73 | 5.74 |
| Overall mFR | 31.90 | **21.85** |

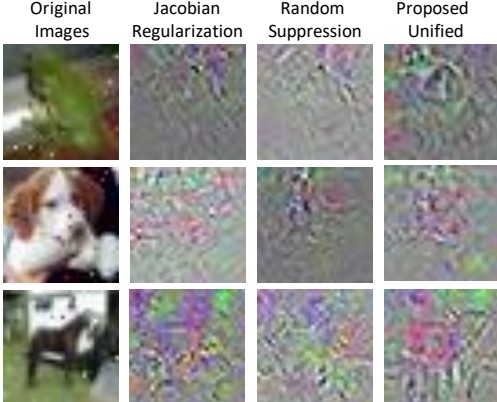

Figure 9: Sample images corrupted with impulse noise and the corresponding dominance matrices obtained using Jacobian regularization, random suppression, and the proposed unified model (Best viewed in color).

Table 6: Comparison of Mean Flip Rate (mFR) of DIVINE with existing algorithms on the CIFAR100-P dataset. A lower mFR indicates better performance.

| Method | mFR % |
|---|---|
| MixUp (Zhang et al., 2018) | 8.9 |
| CutOut (DeVries, 2017) | 11.9 |
| CutMix (Yun et al., 2019) | 12.0 |
| AugMix (Hendrycks et al., 2019) | 5.6 |
| PixMix (Hendrycks et al., 2022) | 5.6 |
| IpMix (Huang et al., 2023) | 4.3 |
| **DIVINE** | **3.1** |

abridge learning. We have also compared the performance of DIVINE with existing methods and the results are shown in Table 6. The proposed algorithm DIVINE outperforms the second-best algorithm IpMix (Huang et al., 2023) by **mFR 1.2%** and achieves the **state-of-the-art (sota)** results.

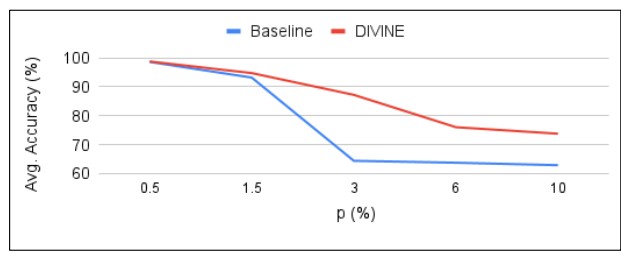

Figure 10: Plot representing the trend line of the average classification accuracy corresponding to baseline and proposed algorithms on different values of p on the MNIST dataset.

Table 7: Asymptotic time and space complexity of DIVINE.

| | Time complexity | Space complexity (GPU) |
|---|---|---|
| DIVINE | $(K\,E_1 + E_2)\,\dfrac{N}{B}\big(F + G + J\big)\ \approx\ 2\,(K\,E_1 + E_2)\,T_{\text{base}}$ | $\mathcal{O}(P + BHWC)$ |

The visualization of the dominance matrix obtained corresponding to the unified model and existing approaches on the corrupted images of the CIFAR10 dataset are shown in Figure 9. We can see that the unified model focuses on multiple input features/regions while the existing approaches fail to focus diversely. On observing the relative corruption error corresponding to impulse noise corruption in Table 2, it is found that the error for the proposed method is 73.02, which is 18.68 and 19.15 less than Jacobian regularization and random suppression methods, respectively. This shows the applicability of the proposed method in real-world scenarios where external corruptions are common. Additionally, the proposed method can be used in combination with existing approaches for improving robustness.

### 5.3 Ablation Experiment to Visualize the Trend of Parameter $p$

We conduct a series of experiments for varying values of $p$, specifically at 0.5%, 1.5%, 3%, 6%, and 10%. The trend observed in average classification accuracy for these values is depicted in Figure 10, and detailed results are presented in Table 1. From this representation, we notice a slight decrease in accuracy at $p$=0.5% and $p$=1.5% for the baseline model, indicating that dominant features are still present in the datasets with feature suppression at these lower $p$ values. However, at $p$=3%, there is a noticeable decline in the baseline model's performance, highlighting the successful elimination of dominant features in the dataset. Consequently, we have selected $p$=3% as the optimal value for conducting our experiments.

### 5.4 Computational Complexity and Runtime

In Table 7, we illustrate the time and space complexity of the proposed DIVINE algorithm. We let $N$ denote the number of training images, $B$ the mini-batch size, and $P$ the total number of model parameters; the quantities $F$, $G$, and $J$ are the per-batch costs of a forward pass, a parameter-gradient backward pass, and an additional Jacobian-through-input backward pass, respectively. The hyper-parameters $K$, $E_1$, and $E_2$ correspond to the number of suppression rounds and the epochs spent in Stages 1 and 2, while $H \times W \times C$ is the spatial resolution of an input image. Using these notations, the table shows that DIVINE's overall training time scales as $2\,(KE_1 + E_2)$ times the baseline cost, with no change in the leading-order memory footprint.

Despite this increase in training-time complexity, the inference-time cost of the unified model trained using DIVINE remains identical to that of standard models, which is a significant practical advantage. This additional computational investment during training enables the model to learn both dominant and inconspicuous features, resulting in improved robustness and generalization. Thus, the added training cost represents a strategic trade-off for greater reliability without affecting deployment efficiency.

Further, we have calculated the temporal cost associated with the MNIST, CIFAR10, CIFAR100, and TinyImageNet datasets during the first phase. In this phase, the model undergoes training on the dataset with suppressed features, and this process is repeated three times. The MNIST dataset requires 10 epochs

Table 8: Training time of Abridged Learning and the proposed DIVINE method on CIFAR10 dataset.

| | Abridged Learning | DIVINE |
|---|---|---|
| Time per Epoch (in seconds) | 73 | 177 |
| Total Training Time (in minutes) | 25 | 60 |

for training, with each epoch taking approximately 14 seconds to complete. Therefore, the entire training process for MNIST takes approximately 420 seconds (7 minutes). In the case of CIFAR10 and CIFAR100 datasets, each epoch lasts about 61 seconds, while for the TinyImageNet dataset, it is around 935 seconds per epoch. These time durations present opportunities for optimization, possibly through the application of methods like the one proposed by (Wang et al., 2020), which involves pruning the network at the initial stage, prior to training. It is important to note that the time required for inference (testing) remains consistent between the proposed unified model and the AL model. Further, we compute the time complexity of Abridge Learning and the proposed DIVINE algorithm for each epoch as well as the total training time. From Table 8 we observe that additional overhead per epoch is 104 seconds, which is mostly spent on the computation of Jacobians. Since the proposed DIVINE method increases the computation time over Abridge Learning, we consider this a limitation of the proposed method the minimization of this overhead can be explored in the future work.

## 6 Limitations

We acknowledge the following limitations of the proposed DIVINE algorithm:

- **Computational Complexity:** Although DIVINE effectively mitigates Abridge Learning, its iterative feature suppression and retraining mechanism significantly increases computational demands. The repeated identification and suppression of dominant features, coupled with subsequent retraining, inherently requires more computational resources and time. However, this iterative approach is crucial for DIVINE's ability to comprehensively learn both dominant and inconspicuous features, substantially improving robustness and generalization. To address this computational limitation, we plan to explore optimization strategies such as gradient approximation methods, selective iteration approaches, adaptive early stopping criteria, and model compression or pruning techniques.

- **Modality and Task Generalizability:** The current validation of DIVINE has been conducted solely within image-based classification tasks. While the underlying principles of DIVINE could be theoretically applicable across diverse modalities and tasks, its practical effectiveness in contexts such as text, audio, or video remains to be empirically verified.

- **Applicability to Non-Semantic Dominant Features:** DIVINE specifically targets semantic dominant features; thus, it is not suitable for addressing Abridge Learning problems driven by non-semantic features, such as color biases observed in datasets like ColoredMNIST. In these scenarios, the spurious correlation arises from attributes lacking semantic meaning, making them difficult to capture using DIVINE's feature dominance approach. Future efforts should focus on integrating complementary bias mitigation techniques, such as specialized feature attribution methods, targeted data augmentation strategies, or domain adaptation frameworks, to effectively handle non-semantic feature biases.

## 7 Conclusion and Future Work

Conventional deep learning methods typically prioritize optimizing classification accuracy, which often leads to overreliance on dominant features and consequently poor generalization to out-of-distribution scenarios. In this paper, we introduced *Diverse and Inconspicuous Feature Learning* (DIVINE), a novel framework explicitly designed to mitigate Abridge Learning by systematically identifying and leveraging diverse inconspicuous yet discriminative features. Extensive empirical evaluations across multiple benchmark datasets, including MNIST, CIFAR10, CIFAR100, TinyImageNet, and their corrupted variants, demonstrate DIVINE's superior

robustness and generalization compared to state-of-the-art baselines. Our dominance maps effectively guide the model in learning comprehensive feature representations, significantly enhancing robustness and ensuring reliable predictions under data corruption and distribution shifts. Future work will focus on extending DIVINE to additional modalities and diverse machine learning tasks, addressing scenarios involving non-semantic dominant features, and optimizing computational efficiency to broaden its applicability in practical, real-world contexts.

## 8 Acknowledgements

This research is supported by a grant from IndiaAI and Meta via the Srijan: Centre of Excellence for Generative AI. Thakral was partially supported by PMRF Fellowship, Mittal was partially supported by IBM Fellowship, and M. Vatsa was partially supported through the Swarnajayanti Fellowship.

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
