# DIVINE: Diverse-Inconspicuous Feature Learning to Mitigate Abridge Learning

**Reviewed on OpenReview:**

## 1 Results on the CelebA dataset

We conducted additional experiments on the CelebA Liu et al. (2015) dataset for the *Blonde Hair* attribute. We evaluated the performance of the Abridge Learning (AL) model, on the original test set $X$ as well as on feature-suppressed test sets $X_{s1}, X_{s2}, X_{s3}$. The dataset $X_{s1}$ contains images with one dominant input feature suppressed per image, while $X_{s2}$ and $X_{s3}$ contain images with two and three dominant features suppressed, respectively. The performance of the AL model noticeably degrades as more dominant features are suppressed. The performance drops from 94.61% on $X$, to 93.11% on $X_{s1}$, 91.72% on $X_{s2}$, and 90.74% on $X_{s3}$. This confirms that models trained via conventional methods rely on dominant input features. In contrast, the unified model trained using our proposed DIVINE algorithm remains robust across all test sets, achieving 94.97%, 93.60%, 92.05%, and 91.28% on $X, X_{s1}, X_{s2}$, and $X_{s3}$, respectively. This demonstrates the effectiveness of DIVINE in learning a more diverse and stable set of predictive features.

Figure 1 compares the Grad-CAM visualizations of the Abridge Learning (AL) model and the proposed DIVINE model for the Blonde Hair attribute classification task. In Figure 1(a) the attention is tightly focused on the top region of the hair, particularly around the central forehead and hairline in case of AL model. However, attention is more distributed, covering both sides of the hair and the lower face. In this case, the AL model relies narrowly on a small, dominant region (shortcut-like focus), while DIVINE uses a broader context, making it more robust. In case of 1(b) AL focuses on non-informative facial areas, possibly relying on background artifacts while in case of DIVINE, attention shifts to hair texture and tone, likely picking up subtle cues of not being blonde. It shows the reliance of AL on biased or spurious features, while DIVINE captures more semantically meaningful signals. In 1(c) the focus is less clear, with attention smeared across the image in case of AL. On the other hand, DIVINE concentrates on the lower hair region, avoiding facial distractions shows better feature localization even in ambiguous cases, suggesting more stable feature learning. In case of Figure 1(d), AL over-focuses on one side of the face and ear area while DIVINE highlights both sides of the hairline, showing better symmetry and contextual understanding.

## References

Ziwei Liu, Ping Luo, Xiaogang Wang, and Xiaoou Tang. Deep learning face attributes in the wild. In *Proceedings of International Conference on Computer Vision (ICCV)*, December 2015.

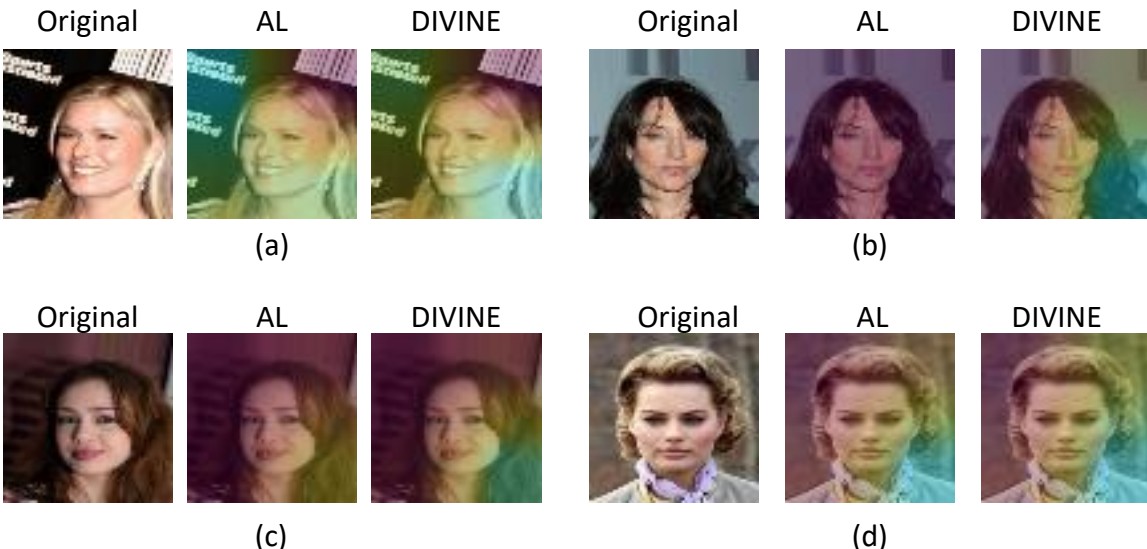

Figure 1: Grad-CAM visualizations for the Blonde Hair attribute classification on the CelebA dataset. Each triplet shows the original image, attention map from the Abridge Learning (AL) model, and attention map from the proposed DIVINE model. (a)–(d) show that while the AL model often over-focuses on dominant or spurious regions, the DIVINE model captures broader and more semantically meaningful features.