# OpenReview forum: "DIVINE: Diverse-Inconspicuous Feature Learning to Mitigate Abridge Learning"
_TMLR — Accepted by TMLR_

### Review · Reviewer_gg3a · 2025-05-13

**Summary Of Contributions:**

This paper addresses an abridged learning problem, in which a model primarily learns prominent features while subtle yet discriminative (inconspicuous) features are ignored, resulting in vulnerability to distribution shifts.
To leverage inconspicuous features, the proposed method DIVINE first identifies dominant features learned by a model and then suppresses them to focus on inconspicuous features.
Experimental results show that DIVINE improved classification accuracy and robustness to image corruption.

**Audience:**

Yes

**Claims And Evidence:**

No

**Requested Changes:**

## Major changes
- For W1: Clarifying the definition of inconspicuous features and the difference from spurious features would be persuasive. Discussing related works of shortcut learning focusing on objects, e.g., [a] and [b], and why the existing works on spurious features cannot handle inconspicuous features would also be convincing.
- For W2: Adding methods for suppressing spurious features discussed in the related work section would be helpful.
- For W3: Experimenting on datasets containing known shortcuts, such as waterbirds and CelebA, or datasets used in other related works, would support the efficacy of learning inconspicuous features. Testing distribution shifts other than image corruption, e.g., DomainNet, PACS, or ImageNet-A/R, would also demonstrate the robustness.
- For W4: Visualizing learned features using explanation methods, such as Grad-CAM, would be beneficial.


[a] Singh et al., Don't Judge an Object by Its Context: Learning to Overcome Contextual Bias, CVPR2020.
[b] Choi et al, Why Can't I Dance in the Mall? Learning to Mitigate Scene Bias in Action Recognition, NeurIPS2019.

## Minor changes
- The use of the term abridged learning is inconsistent. In the introduction, it is described as a problem arising during training, but in experimental sections (e.g., Tables 1 and 4), it is framed as a training strategy. This inconsistency should be addressed for clarity.

**Strengths And Weaknesses:**

## Strengths
- S1: The problem of abridged learning is important, and addressing it would align models with human perception.
- S2: Identifying dominant features via gradient and using them for learning inconspicuous features is intriguing.

## Weaknesses
- W1: The definitions of abridged learning and inconspicuous features are ambiguous. They are also known as shortcut learning and spurious features. Although the difference between shortcut and abridged learning is described in footnote 1, the difference remains vague, particularly due to the unclear definition of inconspicuous features.
- W2: In the experiment, more baselines should be compared.
- W3: Checking if abridged learning is allviated is necessary. While the robustness to image corruptions is examined, the efficacy of learning inconspicuous features is not evaluated. Shortcut learning often stems from correlations between objects and their backgrounds; evaluating robustness to distribution shifts involving such semantics would provide stronger evidence.
- W4: A deeper analysis of the trained models is needed to better understand what features are being learned.

---

> ### Author Response · Authors · 2025-06-07
> **We thank the reviewer for providing their feedback and suggestions. Our response to the reviewer’s questions are provided below:**
>
> ### **Definitions of abridged vs. shortcut; inconspicuous vs. spurious**
> **Inconspicuous features** in our work are semantically meaningful yet under utilized input cues that are relevant to the task but receive little gradient flow because the model over-relies on dominant signals (e.g., an animal’s *ear* rather than its *paw*).
>
> **Spurious features**, as studied in shortcut learning literature (e.g., [a], [b]), are correlated but non-causal cues that models latch onto such as background or scene context and that degrade generalization under distribution shift.
>
> *Inconspicuous features are useful but ignored; spurious features are useless yet exploited.*
>
> Approaches like Singh *et al.* [a] and Choi *et al.* [b] aim to mitigate contextual bias but assume prior knowledge or explicit annotations (e.g., bounding boxes, context labels), which DIVINE does not require. Crucially, methods that suppress spurious cues do not actively promote the discovery of ignored yet useful signals. DIVINE, by contrast, suppresses the currently dominant cues and thereby uncovers and amplifies discriminative features the model had failed to attend to without predefined spurious labels or auxiliary supervision.
>
> ### **More methods for spurious feature suppression**
> We have updated the literature in the revised manuscript by adding the methods for suppressing spurious features.
>
> ### **Experiments on CelebA Dataset**
> As suggested by the reviewer, we ran additional experiments on CelebA for the “Blond Hair” attribute.  The Abridge Learning (AL) model, trained on the original images, is evaluated on four test sets:
>
> * $X$ — original images
> * $X_{s1}$ — each image has one suppressed dominant feature
> * $X_{s2}$ — each image has two suppressed dominant features
> * $X_{s3}$ — each image has three suppressed dominant features
>
> | Dataset | AL model | DIVINE unified model |
> |---------|----------|----------------------|
> | $X$ 	| 94.61 %  | **94.97 %** |
> | $X_{s1}$| 93.11 %  | **93.60 %** |
> | $X_{s2}$| 91.72 %  | **92.05 %** |
> | $X_{s3}$| 90.74 %  | **91.28 %** |
>
> The AL model’s accuracy degrades steadily from 94.61 % on $X$ to 90.74 % on $X_{s3}$ as more dominant cues are removed. In contrast, the unified model trained with DIVINE remains robust, dropping only 94.97 % → 91.28 %. This demonstrates DIVINE’s ability to learn a more diverse and stable set of predictive features even when dominant cues are suppressed.
>
> ### **Grad-CAM visualization of learned features**
> We have included the experimental results of CelebA and corresponding Grad-CAM visualizations in Figure 1 (supplementary file) of the revised manuscript. The Grad-CAM visualizations compare the attention maps of the Abridge Learning (AL) model and the proposed DIVINE model for the Blonde Hair attribute classification task. In all four examples (a–d), the AL model focuses narrowly on dominant or shortcut-like regions such as the central hairline in (a), irrelevant facial areas or background in (b), and uneven patches in (c) and (d) indicating over-reliance on limited features. In contrast, the DIVINE model consistently distributes its attention more broadly and semantically across relevant hair regions, capturing both sides of the head and hair texture even in ambiguous or challenging cases. This demonstrates that DIVINE learns a more diverse and robust set of predictive features.
>
> ### **Terminology consistency**
> In the Introduction section, *Abridge Learning* is described as a phenomenon or failure mode of standard training, where models rely solely on dominant features while ignoring other predictive ones because of biased optimization dynamics (e.g., gradient starvation).   In the Experimental sections (e.g., Tables 1 and 4), we use *Abridge Learning* to denote the baseline training setup that is, a model trained with standard cross-entropy loss without any feature suppression or diversity promoting interventions. This baseline represents the practical outcome of the Abridge Learning phenomenon.
>
> **References**
>
> [a] Singh et al., Don't Judge an Object by Its Context: Learning to Overcome Contextual Bias, CVPR2020.
>
> [b] Choi et al, Why Can't I Dance in the Mall? Learning to Mitigate Scene Bias in Action Recognition, NeurIPS2019.

---

> > ### Comment · Reviewer_gg3a · 2025-06-09
> >
> > I appreciate your detailed explanations, additional experiments, and changes made to the original manuscript.

---

### Review · Reviewer_rBiM · 2025-05-19

**Summary Of Contributions:**

The paper introduces novel approach DIVINE (DIVerse-INconspicuous feature lEarning) to address "Abridge Learning," a phenomenon where deep learning models focus more on dominant features in data and neglect other subtle discriminative features. DIVINE employs a two-stage methodology: first identifying inconspicuous features through iterative dominant feature suppression, then training a unified model incorporating both feature types. Extensive experiments on multiple datasets demonstrate improved robustness to corruptions and perturbations, achieving state-of-the-art results on CIFAR100-P.

**Audience:**

Yes

**Broader Impact Concerns:**

The paper doesn't present any significant ethical concerns.

**Claims And Evidence:**

Yes

**Requested Changes:**

No major revision needed, some minor ones which the authors can consider to improve paper quality I feel is:

* Providing the code is critical for reproducibility. Maybe I missed a github link in this paper, but can the authors more clearly point to code and how to implement this method?
* A more thorough analysis of computational complexity analysis across different datasets would help.
* A more detailed explanation on the limitation regarding non-semantic features using concrete examples can throw more light on the method and its inherent weak points.
* some more strategies could be discussed to help reduce computational overhead.
* a more detailed description of selecting dominance threshold 't' will help improve reproducibility

**Strengths And Weaknesses:**

The paper overall has lot of strong aspects, and some minor weaknesses which I feel are outweighed by the strengths.

Strengths:

* The conceptualization of "Abridge Learning" is clear and made distinct from the more common and general shortcut learning
* The two-stage methodology seems sound and innovative with good technical basis
* The paper also effectively makes use of Jacobian-based computations to identify feature importances.
* Paper nicely evaluates multiple datasets and corruption variations.
* The paper helps us get informative visualizations using features maps, dominance matrices.
* Gives best performance compared to existing methods on robustness to standard perturbations

Weaknesses:

* I feel this method would incur high computation requirements due to multiple runs and Jacobian computations.
* It seems limited to images, and we are not yet sure about how this method performs and translates to other modalities, but this is a minor one as it still provides a very useful technique and basis which can be expanded in the future by others.
* I'm still unclear about how to optimally select parameters like 'p' which seem to be important.
* How does this work apply to non-semantic dominant features like color biases etc, would this still be meaningful?
* The work somewhat lacks theoretical analysis of when and why the method outperforms alternatives.

---

> ### Author Response · Authors · 2025-06-07
> **We thank the reviewer for providing their feedback and suggestions. Our response to the reviewer’s questions are provided below:**
>
> ### **Computational cost**
>
> The proposed method does incur higher computational cost, primarily due to repeated model training on feature-suppressed datasets (typically 3 iterations) and Jacobian computations for every input during each iteration, which are known to be expensive. We acknowledge this as a key limitation. While the total training time increases (e.g., $\,\approx 2.4\times\,$ on CIFAR-10, as shown in Table 8), the inference cost remains unchanged, since only the final unified model is used at test time. To mitigate this overhead in future work, we plan to explore Jacobian approximation techniques, early stopping, feature-selection reuse, and training efficient pruning strategies.
>
> ### **DIVINE on Images and Other Modalities**
> Our current work is focused on the image modality, where spatial feature dominance is naturally defined and visualised. Extending DIVINE to other modalities (e.g., text, audio) would require redefining “dominant features” in a modality-appropriate way (e.g., important tokens in NLP or frequency bands in audio). While this is a limitation of the present work, we regard it as a promising direction: the core idea of iterative feature suppression to reveal overlooked signals is, in principle, modality-agnostic, and we hope future research will adapt it across domains.
>
> ### **Choosing the percentage p of suppressed pixels**
> The parameter $p$ controls the size of the dominant region $F_1(x)$ in each input image.  A very small $p$ may miss important dominant cues, whereas a large $p$ can suppress not only the dominant features but also relevant inconspicuous ones, leading to information loss.  In this work we adopt a data-driven heuristic based on the relative performance drop of a standard model (trained under Abridge Learning) when evaluated on the feature-suppressed datasets $X_{s1}, X_{s2}, X_{s3},\dots$  Empirically, we found $p = 3$% to be optimal:
>
> * Smaller values ($p = 0.5$%, $1.5$%) caused only minor degradation in the baseline model’s accuracy, indicating insufficient suppression.
> * Larger values ($p = 6$%, $10$%) degraded performance for both the baseline and DIVINE models, suggesting over-suppression.
> * $p = 3$% produced a pronounced drop in baseline accuracy, confirming that dominant features were suppressed while the DIVINE model maintained strong performance, showing effective compensation via alternative features.
>
> This analysis is visualised in Figure&nbsp;9 and discussed in section&nbsp;5.3 of the paper.
>
> ### **Non-semantic shortcuts (e.g., colour bias)**
> Our method primarily targets semantically dominant features, spatially localised, interpretable cues. For *non-semantic* biases such as colour shortcuts (which may be distributed across the image or reside in specific colour channels rather than spatial structure), our current Jacobian based pixel suppression may not capture or suppress them effectively. In such cases DIVINE may fail to identify or mitigate these biases. Addressing non-semantic shortcuts would require extending our approach to operate in colour or feature space, for example, using channel-wise or representation-level attribution, which we view as a valuable direction for future work.
>
> ### **Theory for performance gaps**
> Our method is built on the hypothesis that standard training with cross-entropy loss suffers from gradient starvation. The model rapidly learns dominant (easily learnable) features while ignoring other predictive yet less salient ones. Pezeshki *et al.* [1] formalise this through a dynamical-systems analysis, showing that gradient descent naturally favours features with larger gradient magnitude and lower interference. This structural bias yields Abridge Learning.
>
> Our approach tends to outperform standard training when
>
> 1. multiple predictive features exist but one dominates early optimisation,
> 2. the task benefits from feature diversity (robustness to distribution shift, avoidance of spurious cues), and
> 3. standard training under-explores alternative predictive directions.
>
> ### Analytical grounding: synthetic experiment
>
> We examine a multinomial logistic regression task in a linearly separable setting where several features are predictive.
>
> * Gradient descent with cross-entropy learns the direction with the largest signal-to-noise ratio.
> * Suppressing this dominant direction enables recovery of other predictive features.
> * The procedure mimics iterative orthogonalisation, akin to component deflation in ICA.
>
> This simple model provides a concrete justification for why feature suppression encourages broader feature identification—something standard training alone does not achieve.
>
> ### **Code release**
> We have added the github code link in section 4.4 of the revised manuscript.
>
> **References**
>
> [1] Pezeshki M, Kaba O, Bengio Y, Courville AC, Precup D, Lajoie G. Gradient starvation: A learning proclivity in neural networks. Advances in Neural Information Processing Systems. 2021 Dec 6;34:1256-72.

---

> > ### Author Response · Authors · 2025-06-07
> >
> > ### **Complexity analysis of the proposed algorithm**
> > In **Table&nbsp;7** of the revised manuscript we summarise the time and space complexity of the proposed DIVINE algorithm.
> >
> > * $N$ — number of training images
> > * $B$ — mini-batch size
> > * $P$ — total number of model parameters
> > * $F$, $G$, $J$ — per-batch costs of a forward pass, a parameter-gradient backward pass, and an additional Jacobian-through-input backward pass, respectively
> > * $K$ — number of suppression rounds
> > * $E_{1}$, $E_{2}$ — epochs spent in stage&nbsp;1 and stage&nbsp;2
> > * $H \times W \times C$ — spatial resolution and channels of each input image
> >
> > With these symbols, the Table 7 of the revised manuscript shows that DIVINE’s overall training time scales as  $ 2\,(K E_{1} + E_{2}) \times \text{(baseline cost)} $ while the leading-order memory footprint remains unchanged. We have updated the runtime complexity in section 5.4 of the updated manuscript.
> >
> >
> > ### **Limitation examples**
> > In datasets like ColoredMNIST, where the label is spuriously correlated with a specific color channel, the dominant feature is not spatial or semantic but global and statistical (e.g., background color). Since our method relies on pixel-wise Jacobian magnitudes, it is not well suited to detect or suppress such distributed or channel-level biases.
> >
> >
> > ### **Strategies to Reduce Overhead**
> > In addition to the optimisations already discussed, we outline several strategies to further reduce computational overhead:
> >
> > * **Jacobian approximation:**  Replace full Jacobian computations with efficient gradient approximations, such as finite differences or randomised projections.
> > * **Feature reuse:** Cache and reuse dominance maps once feature dominance stabilises across epochs, avoiding redundant computations.
> > * **Early stopping in iterative suppression:** Dynamically halt suppression rounds when sufficient features have been identified (e.g., when validation performance plateaus), instead of using a fixed number of iterations.
> > * **Subset sampling:** Apply suppression and Jacobian computation to a representative subset of the data rather than the entire training set.
> > * **Multi-feature identification in one pass:** Investigate clustering or attribution methods that can surface several features within a single training round.
> >
> > These approaches are discussed in section&nbsp;7 (Conclusion and Future Work) of the revised manuscript.
> >
> > ### **Towards Dominance threshold t**
> > In our implementation, for each input image we sort the absolute entries of the dominance matrix $|D_{1}(x)|$ in descending order and select the top $p\%$ of pixels. The threshold $t$ is set to the value at the $p$-th percentile of this sorted list. All pixels satisfying $|D_{1}(x)| \ge t$ are retained in the binary mask $M_{1}(x)$.
> >
> > **References**
> >
> > [1] Pezeshki M, Kaba O, Bengio Y, Courville AC, Precup D, Lajoie G. Gradient starvation: A learning proclivity in neural networks. Advances in Neural Information Processing Systems. 2021 Dec 6;34:1256-72.

---

> > > ### Comment · Reviewer_rBiM · 2025-06-08
> > > **The authors have addressed my concerns**
> > >
> > > Thank you that addresses most of my concerns reasonably well. I appreciate your efforts and the changes you've made to the original manuscript.

---

### Review · Reviewer_nDyE · 2025-05-24

**Summary Of Contributions:**

This work studies the phenomenon called Abridge Learning or shortcuts, when supervised learning methods primary pick out a dominant feature that impact the training loss, and overlook other important features that are semantically relavant, hence resulting poor out-of-distribution generalization. The author proposed a pipline that enforces the model to also identify other inconspicuous features to enhance generalizability.

**Audience:**

Yes

**Broader Impact Concerns:**

This work focuses on improving learning more diverse features of supervised learning. The work mostly focusing on methodological design. I do not forsee potential concerns on misuse.

**Claims And Evidence:**

No

**Requested Changes:**

Please see the discussion on weakness.

**Strengths And Weaknesses:**

## Strength:
- the motivation of work is clearly stated.
- the methodology is fairly simple and intuitive to understand.

## Weakness:
- This work lack a careful mathematical formulation on the Abridge Learning phenomenon.  I would recommend having a precise mathematical definition of the AL. For example, assume the data distribution follows a structure causal model (SCM), can you formulate AL learning precisely using rigorous identifiability-type argument?

- The author discusseed a few prior works related to the Abridge Learning phenomenon, but to my knowledge, I don't think there is a consensus in literature that the cross entropy loss will "identify" the most dominant feature, particularly given that there is no precise identifiability argument supporting this. I would like to see rigorous theoretical discussion on this phenomenon, supported with an illustrating simulation example, given that the AL phenomenon serves as the main motivation of this work.

-  Sec 3.1 - 3.2 deserves some clarification on the notations. Specifically, "feature learned using the loss function Eq.1", what does this mean? There is no $F$ in Eq.1. Also the input feature $F_i$ (first sentence in Sec 3.1) is not defined (does the definitnion of $F_i$ follows Eq(5)?) Also, for preciseness, I recommend the authors specify explicitly the input and output spaces/dimensions of the feature map, mask, etc.

-  I find the notion "the identified feature" (right above Eq(6)) very confusing. Does $F_1(x)$  correspond to a subset of pixels? From Eq(5) it looks like the feature just represents a subspace of the original input data? (this is very different from the typical language used in representation learning).

- From the cross entropy loss (Eq1.), it seems to be more natural to perform Tayloe expansion on $\log f$ instead of $f$, and define the dominance matrix around the Jacobian of $\log f$. I'm curious whether it's a viable option.

-  The Jacobian matrix is in some sense affine equivarient, meaning that I can alter the relative scale of the Jacobian matrix values by apply affine transformations to the raw input X. With different pre-processing applied to the same original dataset, the dominance map may identify completely different features. Can the author elaborate on this?

- I'm not convinced that the proposed methodology achieved the authors original goal---leaning more diverse features instead of just the dominance one. Most of the experimental results focus on looking the classification quality of the methods on corrupted test set, but not examining the learned features/latent representations. I'd like to see both a mathematical justification of this methodology in the setting of simple multinomial logistric regression (let's assume the true model is going to be linear) and an empirical validation. I think this work will benefit from having a proper identification results (even in the linear setting, mirroring the standard (non)linear ICA-type of argument, e.g., **Hyvarinen, A. and Pajunen, P. Nonlinear independent component analysis: Existence and uniqueness results.**)

- Finally, I think this paper lacks a careful discussion on literature around causal representation learning, disentanglement. For example, the set of identifibility analysis on ICA mentioned above, some recent advancedment in identifiabilty on generative models (**Indeterminacy in Generative Models: Characterization and Strong Identifiability**) , and additive identifiability  (**Additive Decoders for Latent Variables Identification and Cartesian-Product Extrapolation**).

---

> ### Author Response · Authors · 2025-06-07
> **We thank the reviewer for providing their feedback and suggestions. Our response to the reviewer’s questions are provided below:**
>
> ### **Adding a rigorous mathematical formulation for Abridge Learning**
> We have added a mathematical formulation of the proposed algorithm. It is added in section 1 of the revised manuscript and also included here for your ready reference.
>
> Consider an observed input image $x \in \mathbb{R}^{H \times W \times C}$ generated via a Structural Causal Model (SCM):
>
> $$
> x = f_X(Z), \quad y = f_Y(Z_S),
> $$
>
> where $Z \in \mathbb{R}^{k}$ represents latent generative factors tied to semantically meaningful features, and $Z_S \subseteq Z$ indicates the subset causally influencing the label $y \in \mathcal{Y}$. Given a predictive model $f(x; \theta)$ parameterized by $\theta$, trained via empirical risk minimization using cross-entropy loss:
>
> $$
> \min_{\theta} \mathbb{E}_{x,y}\left[-y^\top \log f(x; \theta)\right],
> $$
>
> we formally characterize **Abridge Learning** as the condition when the learned predictor $f(x; \theta^*)$ predominantly exploits dominant features $Z_D \subseteq Z_S$, identified by strong gradient signals and high mutual information with labels, while neglecting inconspicuous yet predictive features $Z_I = Z_S \setminus Z_D$. Mathematically, this phenomenon can be stated as:
>
> $$ I(f(x; \theta^{\star}); Z_D) \gg I(f(x; \theta^{\star}); Z_I), \quad \text{with} \quad I(Z_I; Y \mid Z_D) > 0. $$
>
> Consequently, the reliance on dominant features results in significantly degraded performance under distributional shifts or feature perturbations:
>
>
> $$
> \mathbb{E}_{x', y'}[\mathcal{L}(f(x'; \theta^{\]star}), y')] \gg \mathbb{E}_x, {}_y [ [\mathcal{L}(f(x; \theta^{\star}), y)] ,
> $$
>
>
> where $x'$ denotes inputs with corrupted or missing dominant features. To address this challenge, we propose a novel method that explicitly suppresses overly exploited dominant signals, allowing diverse inconspicuous features to emerge and contribute meaningfully to classification. By enriching the learned representations with these subtle yet causal cues, our method significantly enhances robustness and generalization across varied data distributions.
>
>
> ### **Theoretical Discussion and Simulation**
>
> Pezeshki et al. [1] rigorously show that when a neural network is trained with cross-entropy loss, gradient dynamics naturally prioritize features with higher alignment to the label signal and lower interference. This results in a feature imbalance, where early learned features suppress gradients flowing into less dominant but still predictive features. They term this phenomenon as gradient starvation. They demonstrate that gradient descent leads to convergence on a subset of features even when multiple informative ones are available. This provides precise theoretical backing for the Abridge Learning phenomenon. The optimization landscape under cross-entropy loss does not favor full feature recovery, but rather rapid convergence to the most easily learnable (dominant) features.
>
> **Experimental Validation**: To support the above claim, we conducted a synthetic data experiment with two features, $z_1$ and $z_2$, both of which are informative. As shown in Figure 3 of the revised manuscript, under standard training with cross entropy loss, the model learns to rely on $z_2$, and suppressing $z_2$ degrades model performance unless retrained, thus illustrating **Abridge Learning**. We also showed how the proposed algorithm is able to mitigate this problem by utilizing **both** $z_1$ and $z_2$.
>
> * **Abridge Learning**: In the case of Abridge Learning, the decision boundary is **horizontal**, separating classes primarily based on the $z_2$ axis. This indicates that the model is relying **entirely on $z_2$** to make predictions. This supports the Abridge Learning phenomenon — i.e., the model under-utilizes other predictive features such as $z_1$ once it finds an easy solution with $z_2$.
>
> * **Identify and Learn Inconspicuous Feature**: With $z_2$ suppressed (masked), the model is forced to rely on the previously underutilized feature $z_1$. This demonstrates that $z_1$ was a valid predictive feature, but was ignored during Abridge Learning due to the optimization dynamics of cross-entropy. It is recoverable if we explicitly remove the dominant feature, confirming that Abridge Learning hides useful information during standard training.
>
> * **Unified Learning**: In this setting, the model has successfully learned to combine both features for decision making, as the decision boundary is diagonal, showing influence from both $z_1$ and $z_2$. This illustrates that the model overcomes the optimization bias of cross-entropy toward dominant features, resulting in more complete feature utilization and improved robustness and generalization.
>
> We have included this synthetic data experiment in section 3 of the revised manuscript.
>
>
> **References**
>
> [1] Pezeshki M, Kaba O, Bengio Y, Courville AC, Precup D, Lajoie G. Gradient starvation: A learning proclivity in neural networks. Advances in Neural Information Processing Systems. 2021 Dec 6;34:1256-72.

---

> > ### Author Response · Authors · 2025-06-07
> >
> > ### **Refining the Notations in Sec.​3.1 and 3.2**
> >
> > We have updated Sections 3.1 and 3.2 in the revised manuscript with additional clarifications on the notations and explicitly specified the dimensions of the input, feature map, mask, etc. We have included the definition of $F_i(x)$ here for reference:
> >
> > Let $F$ denote the set of input features corresponding to masked versions of the image, where each $F_i(x) \in F$ is a masked variant of the input image $x \in \mathbb{R}^{H \times W \times C}$ and has the same dimensionality $H \times W \times C$. Each $F_i(x)$ highlights a subset of pixels associated with input features learned by the model. These highlighted regions typically correspond to features that are salient or informative under standard predictive models.
> >
> > ### **“Identified feature” terminology**
> > * In our paper, the term $F_1(x)$, defined in Eq. (5), indeed corresponds to a subset of the input image pixels, not a latent or abstract learned representation. Specifically, $F_1(x) = M_1(x) \odot x$. Here, $M_1(x) \in \{0, 1\}^{H \times W \times C}$ is a binary mask that selects the most dominant input pixels based on the Jacobian derived dominance matrix $D_1(x)$. Thus, $F_1(x)$ can be viewed as a masked version of the original input $x$, retaining only the pixel regions the model deems most influential for classification.
> >
> > * We agree that this is different from typical “features” in representation learning. In our case, we are referring to input space features, i.e., spatially localized regions in the input image. These regions are used as proxies for dominant semantic cues, identified via input gradients.
> >
> >
> > ### **Building the dominance matrix on log f instead of f**
> >
> > Thank you for this insightful suggestion. Indeed, since the cross-entropy loss is defined as
> > $\mathcal{L}(x, y) = -y^\top \log f(x; \theta)$, it is entirely reasonable to consider the Taylor expansion of $\log f(x; \theta)$ rather than $f(x; \theta)$ itself, especially when we are interested in how small perturbations in the input $x$ influence the loss directly. In fact, if we denote $\ell(x) := \log f(x; \theta)$, then the first-order Taylor expansion yields $\ell(x + \epsilon) \approx \ell(x) + \epsilon^\top \nabla_x \log f(x; \theta),$ and hence, the sensitivity of the loss to input perturbations is governed by the Jacobian $\frac{\partial \log f(x)}{\partial x}$, particularly for the true class dimension.
> >
> > This log Jacobian based dominance matrix could serve as an alternative way to compute pixel-wise attribution or dominance. It may have benefits such as:
> >
> > - Tighter alignment with the loss surface.
> > - Better sensitivity modeling for low-probability classes where gradients of $f(x)$ may vanish, but $\log f(x)$ still has meaningful gradient flow.
> >
> > However, we chose to base our dominance measure on $\frac{\partial f(x)}{\partial x}$ because of the following reasons:
> >
> > - This formulation aligns with existing literature on input saliency and gradient-based attribution methods (e.g., saliency maps, Grad-CAM, etc.).
> > - In practice, gradients of $f(x)$ already reflect the sensitivity of the model’s confidence scores to input perturbations, which is particularly useful when suppressing dominant classification cues.
> >
> > That said, the suggestion is viable and potentially valuable. In future work, we plan to explore whether the log softmax Jacobian provides improved localization of dominant input regions or better supports the suppression process, especially in low-confidence settings or multi-class ambiguities.
> >
> > ### **Affine equivariance of the Jacobian**
> >
> > The Jacobian $\frac{\partial f(x)}{\partial x}$, being a first-order derivative of the model output with respect to the input, is sensitive to affine transformations of the input space. Specifically, for an affine transformation $x' = A x + b$, the chain rule implies:
> >
> > $$
> > \frac{\partial f(x')}{\partial x'} = \frac{\partial f(Ax + b)}{\partial x} A
> > $$
> >
> > which shows that the Jacobian transforms linearly with the input. Consequently, the dominance matrix, which is computed from the Jacobian, is not invariant to preprocessing steps like input rescaling, normalization shifts, or contrast adjustments. This means that different preprocessing pipelines (even standard ones like per-channel normalization or histogram equalization) can yield **different dominance maps**, and thus different identified features, potentially affecting the set of "dominant" and "inconspicuous" features selected during suppression.

---

> > > ### Author Response · Authors · 2025-06-07
> > >
> > > ### **Formal and empirical validation of diversity gain**
> > >
> > > To formalise the intuition, consider a multinomial logistic regression model.  Let $x \in \mathbb{R}^{d}$, $y \in \{1,\dots,K\}$, and assume the true model is linear:
> > >
> > > $$
> > > P(y=k \mid x)=\frac{\exp\!\bigl(w_{k}^{\top}x\bigr)}
> > >                	{\sum_{j=1}^{K} \exp\!\bigl(w_{j}^{\top}x\bigr)}.
> > > $$
> > >
> > > Suppose $x = A z$, where $z \in \mathbb{R}^{r}$ are latent generative features (*e.g.*, semantic factors) and $A \in \mathbb{R}^{d \times r}$ is a mixing matrix.
> > > Let the true discriminative subspace be spanned by a sparse set of $z$ dimensions, say $z_{D} \subset z$. Then
> > >
> > > $$
> > > P(y=k \mid z)=\frac{\exp\!\bigl(w_{k}^{\top} A z\bigr)}
> > >                	{\sum_{j=1}^{K} \exp\!\bigl(w_{j}^{\top} A z\bigr)}.
> > > $$
> > >
> > > * During standard training, gradient flow concentrates on the coordinates in $z_{D}$ that yield maximal discriminative power early on. Features orthogonal to these, while still potentially predictive receive starved gradients [1], leading to what we term Abridge Learning.
> > >
> > > * Our iterative feature-suppression mechanism alters this dynamic: masking dominant contributions in $x$ effectively projects out the current principal direction, forcing the model to rely on alternative directions in input space (corresponding to other components of $z$).
> > > * In the linear setting this is analogous to successive deflation in ICA, where identified components are removed to isolate the remainder.
> > >
> > > Under these assumptions (linearly independent latent features, each contributing additively to class separation) we can state an identifiability condition: successively suppressing the top-contributing directions will eventually reveal the remaining predictive components, provided each carries non-zero signal.
> > >
> > > We validated the claim with a synthetic two-feature experiment ($z_{1}$, $z_{2}$).  Figure&nbsp;3 of the revised manuscript shows that standard training converges on $z_{2}$; masking $z_{2}$ degrades performance unless the model is retrained, illustrating Abridge Learning.  Conversely, our algorithm successfully recovers and utilises both $z_{1}$ and $z_{2}$, mitigating the optimization bias toward dominant features.
> > >
> > > ### **Discussion on Causal Representation Learning and Disentanglement**
> > >
> > > Our motivation aligns with the goal of causal representation learning, identifying semantically meaningful and (ideally) independent generative factors that influence observed outcomes.  Unlike most disentanglement approaches that attempt to learn the full latent space directly through generative models such as VAEs or GANs, our method is purely discriminative as it exposes new predictive factors by suppressing previously learned ones. Although we make no explicit causal graph assumption, the resulting iterative suppression acts like successive latent factor identification, naturally connecting our work to recent results on identifiability.
> > >
> > >
> > > ### Relevant identifiability literature
> > >
> > > * **Indeterminacy in Generative Models**: Shows that typical generative models ($\,$VAEs, GANs$\,$) suffer from *structural indeterminacy* unless strong identifiability conditions are imposed.  Our approach sidesteps this limitation by not relying on latent generation, while sharing the same goal of interpretable, causal feature discovery.
> > >
> > > * **Additive Decoders for Latent Variable Identification:** Demonstrates that additive decoder structures yield identifiable latent components. Our iterative masking–retraining scheme is analogous: suppressing one *additive* contribution (the dominant feature) enables the model to discover orthogonal or independent components.
> > >
> > > We have incorporated these works into the Related Work section of the revised manuscript.
> > >
> > > **References**
> > >
> > > [1] Pezeshki M, Kaba O, Bengio Y, Courville AC, Precup D, Lajoie G. Gradient starvation: A learning proclivity in neural networks. Advances in Neural Information Processing Systems. 2021 Dec 6;34:1256-72.

---

### Decision · Action_Editor_ePCt · 2025-06-20

**Recommendation:** Accept with minor revision

**Additional Comments:**

The Grad-CAM visualizations in the supplementary material could benefit from clearer presentation, as the current results make it challenging to assess whether the model captures subtle yet semantically meaningful features. Evaluating DIVINE on datasets with semantical distribution shifts would further strengthen the claim that these inconspicuous features contribute to improved generalization.

**Audience:**

Yes

**Audience Explanation:**

The proposed DIVINE framework offers a novel perspective on addressing this issue by explicitly identifying and incorporating inconspicuous but semantically relevant features. Given the growing interest in model robustness, interpretability, and out-of-distribution generalization, researchers working in these areas would find the ideas and empirical insights of this paper relevant.

**Claims And Evidence:**

Yes

**Claims Explanation:**

The submission presents a clear and well-motivated approach to addressing the issue of "Abridge Learning" by encouraging the model to attend to both dominant and inconspicuous features. The proposed method, DIVINE, is described in sufficient detail, and the experimental setup covers multiple datasets, including tests under distribution shift and corruption scenarios. These results provide evidence that DIVINE improves model robustness and outperforms baselines in certain settings.